# Mesh Based Simulations with Spatial and Temporal awareness

Paul Garnier [1]   Vincent Lannelongue [1]   Elie Hachem [1]

## Abstract

Machine Learning surrogates for Computational Fluid Dynamics (CFD), particularly Graph Neural Networks (GNNs) and Transformers, have become a new important approach for accelerating physics simulations. However, we identify a critical bottleneck in the field: while architectures have advanced significantly, the common underlying training paradigms remain bound to naive assumptions, such as node-wise supervision and explicit Euler time-stepping. These legacy choices ignore the stiff dynamics and local flux continuity inherent to numerous partial differential equations resolution methods, such as Finite Element, Difference, or Volume (FEM). In this work, we propose a unified framework to bridge the gap between geometric deep learning and rigorous numerical analysis. We introduce three key innovations: (1) Multi Node Prediction, a stencil-level objective that predicts field values for a node's full local topology, enforcing spatial derivative consistency; (2) Temporal Correction, replacing unstable explicit schemes with a predictor–corrector via temporal Cross-Attention; and (3) Geometric Inductive Biases, leveraging 3D Rotary Positional Embeddings (RoPE) to robustly capture rotational symmetries in unstructured meshes. We evaluate this framework across three architectures (MeshGraphNet, Transolver, and a Transformer) on diverse physics datasets. Our approach yields consistent improvements in accuracy and stability, particularly in long-horizon rollouts, while producing latent representations that generalize to unseen subtasks such as Wall Shear Stress or Pressure prediction. Code is available at https://github.com/DonsetPG/graph-physics.

*Equal contribution [1]CEMEF - Mines Paris PSL. Correspondence to: Paul Garnier <paul.garnier@minesparis.psl.eu>.

*Proceedings of the $43^{rd}$ International Conference on Machine Learning*, Seoul, South Korea. PMLR 306, 2026. Copyright 2026 by the author(s).

## 1. Introduction

Simulating physical phenomena, particularly CFD, consist of solving Partial Differential Equations (PDEs) over complex geometries discretized as unstructured meshes. While traditional solvers are made of large-scale linear algebra kernels, they suffer from a fundamental inefficiency: every new simulation starts from scratch, ignoring the potentially useful data from previous runs. This computational bottleneck has motivated the rapid adoption of Machine Learning (ML) as a reusable surrogate for physics simulation.

Early data-driven approaches targeted structured grids using Convolutional Neural Networks (CNNs) (Thuerey et al., 2018) or Physics-Informed Neural Networks (PINNs) (Raissi et al., 2019). However, the geometric complexity of real-world engineering necessitates unstructured discretizations. Notably, Graph Neural Networks (GNNs) and Message-Passing Systems (MPS) (Battaglia et al., 2018; Pfaff et al., 2021) have emerged as the dominant paradigm, enabling predictions on irregular domains ranging from weather forecasting (Lam et al., 2023) to hemodynamics (Suk et al., 2024). To handle scaling, recent works have integrated FEM-inspired multigrid strategies (Fortunato et al., 2022; Garnier et al., 2024) or adopted Transformer-based architectures capable of global attention (Wu et al., 2024).

Despite these architectural advances, the fundamental *training methodology* has remained largely unchanged since the creation of MeshGraphNet. We argue that the community has over-optimized network architectures while neglecting the numerical priors required to solve PDEs effectively. We identify two specific failures in the current state-of-the-art:

1. **Node-wise prediction:** Standard losses minimize error per node in isolation. In FEM, however, physics is defined by the flux across element boundaries. Predicting a single node ignores the local differential information essential for conservation laws.

2. **Explicit Euler scheme:** Many common discrete-time surrogates update the state via $\mathbf{u}_{t+\Delta t} = \mathbf{u}_t + \Delta t \Phi(\mathbf{u}_t)$. This mimics an explicit Euler scheme, which can be numerically unstable for stiff dynamics and prone to error accumulation over long trajectories.

In this paper, we challenge these legacy choices. We propose

a methodology that aligns ML surrogates with the principles of rigorous numerical solvers. We argue that a surrogate must predict not just the value at a point, but the local stencil of the solution, and that temporal evolution requires attention-based integration to handle stiff timesteps. We present the different contributions of this paper below.

- Multi Node Prediction: We introduce a training task where the model predicts the next state for a node *and* its topological neighbors. This acts as a regularizer that enforces local smoothness and continuity, similar to flux reconstruction in FEM.

- Temporal Correction: We replace the explicit Euler residual connection with a multi-step Cross-Attention mechanism. This allows the model to approximate an implicit time-stepping scheme by leveraging spatial predictions, thereby significantly improving stability.

- Complete Ablation study: We evaluate multiple improvements of standard architecture, including 3D Rotary Positional Embeddings, attention variants, activation function, and loss objectives. We present them in orange.

- Model-Agnostic Evaluations: We validate these contributions across three distinct architectures (MeshGraphNet, Transolver, Transformer) and three primary datasets, with additional large-scale, long-range, and geometry-shift stress tests, demonstrating that our improvements scale across both model size and physical complexity.

## 1.1. Related Work

**Supervsied loss for GNNs**    While a large amount of methods now improve the supervised loss using Physics Informed losses (Cai et al., 2021; Liu et al., 2025), we aim our focus towards supervised tasks. (Belkin et al., 2006; Zhang & Chen, 2018) defined supervised loss where the model is trained to predict properties of adjacent nodes (e.g., edge existence or edge labels) using losses defined over $(u, v) \in E$, often implemented as link prediction or pairwise classification with negative sampling. A closely related approach (Alsentzer et al., 2020) reframes node learning as subgraph learning: each labeled node induces a 1-hop enclosing subgraph; the GNN produces a representation of the induced neighborhood via pooling; and supervision is applied at the subgraph level rather than directly on the center node embedding. However, most of those approaches are unfit for node regression tasks. A closely related body of work is the study of multiple-token predictions in NLP. (Gloeckle et al., 2024) introduced a supervised task to predict multiple future tokens, thereby shifting learning from single-instance targets to richer, structured targets that better capture local

context and dependencies. Later on, a similar approach was used by DeepSeek in (DeepSeek-AI et al., 2025).

**Temporal Schemes for Machine Learning surrogate**    In many classical setups, one can decouple the learned spatial operator $f_\theta$ (or learned increment $\Delta u_\theta$) from the temporal integrator, and then advance the state with higher-order explicit schemes (e.g., midpoint (Butcher, 2016), Runge-Kutta2 (RK2) (Runge, 1895; Kutta, 1901), RK3/RK4, or multistep Adams-Bashforth (Hairer et al., 1993)) by evaluating the surrogate multiple times per step. Several learned simulators also depart from residual next-step updates through continuous-time or solver-coupled formulations. Finite Element Networks, for instance, derive a continuous-time model from FEM and estimate latent dynamics on mesh cells before advancing them with a finite-element mass matrix (Lienen & Günnemann, 2022).

**Long-range interactions on meshes**    Oversquashing and long-range dependency bottlenecks have also been studied directly in mesh simulators. PIORF rewires mesh graphs using physics-informed Ollivier-Ricci flow to connect bottleneck regions with high-gradient nodes (Yu et al., 2025); Hamiltonian-based graph simulators target information preservation during long-range propagation (Hoang et al., 2026); and HCMT uses hierarchical contact transformers to propagate collision effects across distant flexible-body regions (Yu et al., 2024). EAGLE further stresses long-range turbulent dynamics with moving sources and varying geometries (Janny et al., 2023).

**Positional Encoding**    For 2D imagery, absolute positional encodings adapt 1D sinusoidal features to a grid by factorizing row and column coordinates and either summing or concatenating the resulting embeddings. Many Vision Transformers (ViTs) adopt learned absolute embeddings over patches (Dosovitskiy et al., 2021), which can be effective but may generalize poorly across resolutions without interpolation. In contrast, relative positional encodings represent pairwise offsets between tokens (e.g., $\Delta x, \Delta y$), enabling attention weights to depend on relative geometry rather than absolute location (Shaw et al., 2018; Wu et al., 2021). In the case of points clouds, a common strategy injects coordinates directly by passing $(x, y)$ or $(x, y, z)$ through an MLP and adding/concatenating the result to point features (Wu et al., 2022). However, raw absolute coordinates can entangle pose with content; thus, many approaches emphasize *relative* or *local* encodings constructed from neighborhoods (e.g., $p_j - p_i$ for neighboring points), distances, and angles (Zhao et al., 2021). Regular positional encoding (Vaswani et al., 2017) has also been adapted to 3D points and graphs (Alkin et al., 2025). In the case of graphs, one can represent positional encoding with Laplacian Eigen Vectors (Dwivedi & Bresson, 2021), learnable positional Encoding (Kreuzer

et al., 2021), and RandomWalk (Dwivedi et al., 2022).

## 2. Preliminaries

### 2.1. Mesh as Graph

We consider a mesh as an undirected graph $\mathcal{G} = (\mathcal{V}, \mathcal{E})$. $\mathcal{V} = \{\mathbf{x}_i\}_{i=1:N}$ is the set of vertices or nodes, where each $\mathbf{x}_i \in \mathbb{R}^p$ represents the attributes of node $i$. $\mathcal{E} = \{(\mathbf{e}_k, r_k, s_k)\}_{k=1:N^e}$ is the set of edges, where each $\mathbf{e}_k$ represents the attributes of edge $k$, $r_k$ is the index of the receiver node, and $s_k$ is the index of the sender node.

For certain models (Transformer and Transolver), we omit edge attributes and treat each node as a token. We note $\mathbf{X} = (\mathbf{x}_1, \mathbf{x}_2, ..., \mathbf{x}_N)^\top \in \mathbb{R}^{N \times p}$ our input matrix, made of $N$ tokens of dimension $p$. Let $\mathbf{Z} = (\mathbf{z}_1, \mathbf{z}_2, ..., \mathbf{z}_N)^\top \in \mathbb{R}^{N \times d}$ be the $d$-dimensional embedding of our nodes. We define $\mathbf{A}$ as the adjacency matrix of our graph, setting $\mathbf{A}_{ij} = \mathbf{A}_{ji} = 1$ if $(i, j) \in \mathcal{E}$ and 0 otherwise.

### 2.2. Models

In the remainder of the paper, we study three different models: MeshGraphNet (Pfaff et al., 2021), Transformer (Garnier et al., 2025b), and Transolver (Wu et al., 2024). All models follow an Encode-Process-Decode architecture similar to (Sanchez-Gonzalez et al., 2020). The Encoder maps the input nodes into a latent space. We then apply $L$ layers of a processor. Finally, the Decoder maps back our outputs into a meaningful space. At each step, our models are autoregressive, meaning that their output is used as input for the next step of the simulation. In the remainder of the section, we briefly present how Encoders and Decoders are built before introducing the basic concepts of our three models.

**Encoder and Decoder**  We use a simple two-layer MLP to encode our inputs. Our encoder $\mathcal{E} : \mathbb{R}^p \to \mathbb{R}^d$ maps our nodes (resp. our edges) $\mathbf{X} \in \mathbb{R}^{N \times p}$ into a latent space $\mathbf{Z} \in \mathbb{R}^{N \times d}$. The parameter $d$ is shared across all our layers as the main width parameter. The Decoder $\mathcal{D} : \mathbb{R}^d \to \mathbb{R}^{2 \text{ or } 3}$ generates a physical output (in 2 or 3D) from the latent space using two layers as well.

**MeshGraphNet**  Recall that each spatial processor in MeshGraphNet are message-passing layers that update both the node and edge attributes given the current node and edge attributes:

$$
\begin{aligned}
\mathbf{e}'_k &= f^e(\mathbf{e}_k, \mathbf{z}_{r_k}, \mathbf{z}_{s_k}) && \forall k \in E \\
\bar{\mathbf{e}}'_r &= \sum_{e \in E'_r} e && \forall r \in V \\
\tilde{\mathbf{z}}_r &= [\mathbf{z}_r, \bar{\mathbf{e}}'_r] && \forall r \in V \\
\mathbf{z}'_r &= f^v(\tilde{\mathbf{z}}_r) && \forall r \in V
\end{aligned}
\tag{1}
$$

where $f^e$ and $f^v$ are simple Multi-Layer Perceptron (MLP).

**Transformer**  Each spatial processor from Transformer is defined using $\text{MHA}(\mathbf{Z}, \mathbf{A}) = \left( \text{softmax}\left( \frac{\mathbf{A} \odot (QK^T)}{\sqrt{d}} \right) \right) V$ where $Q, K, V$ are linear projections of $\mathbf{Z}$ and $\odot$ is the Hadamard product. In our Transformer baseline, $\mathbf{A}$ also includes random long-distance edges, or jumpers, to support nonlocal information exchange; graph-rewiring strategies such as PIORF could be combined with the proposed modules. The overall architecture is then defined as:

$$
\mathbf{Z}'_l = \text{RMSNorm}\big(\text{MHA}(\mathbf{Z}_{l-1}, \mathbf{A}) + \mathbf{Z}_{l-1}\big) \quad \ell \in [1 \dots L]
\tag{2}
$$

$$
\mathbf{Z}_l = \text{RMSNorm}\big(\text{GatedMLP}(\mathbf{Z}'_l) + \mathbf{Z}'_l\big) \quad \ell \in [1 \dots L]
\tag{3}
$$

$$
\tag{4}
$$

where GatedMLP is a Gated Multi-Layer Perceptron (Dauphin et al., 2017).

**Transolver**  The spatial processors from Transolver learn $M$ latent slices from the $N$ vertices of the mesh and process them using a Physics-Attention operator. The processor then broadcasts the information back to the mesh. The processors can be summarized as follows:

$$
\mathbf{z}_j = \frac{\sum_{i=1}^N \mathbf{w}_{i,j} \mathbf{x}_i}{\sum_{i=1}^N \mathbf{w}_{i,j}}, \quad \text{with} \quad \mathbf{w} = \text{Softmax}(\text{MLP}(\mathbf{x}))
\tag{5}
$$

$$
\mathbf{x}'_i = \sum_{j=1}^M \mathbf{w}_{i,j} \cdot \text{Attention}(\mathbf{z})_j
\tag{6}
$$

*Remark* 2.1. The last two architectures are based on an attention mechanism. We also investigate in the appendix several modifications to regular attention: Multi Head Latent Attention (Liu et al., 2024) and variants of Gated Delta Nets and Linear Attention (Qiu et al., 2025; KimiTeam et al., 2025). We add those modifications to our ablation study.

## 3. Methodology

### 3.1. Multi Node Prediction

Recall that many common approaches use a regression task on a per-node basis, where $\mathbf{z}_i^\mathbf{L}$ represents the embedding of node $i$ after $L$ layers:

$$
\mathcal{L}_{\text{main}}(\theta) = \frac{1}{N} \sum_{i=1}^N \ell(\mathcal{D}(\mathbf{z}_i^\mathbf{L}), y_i)
\tag{7}
$$

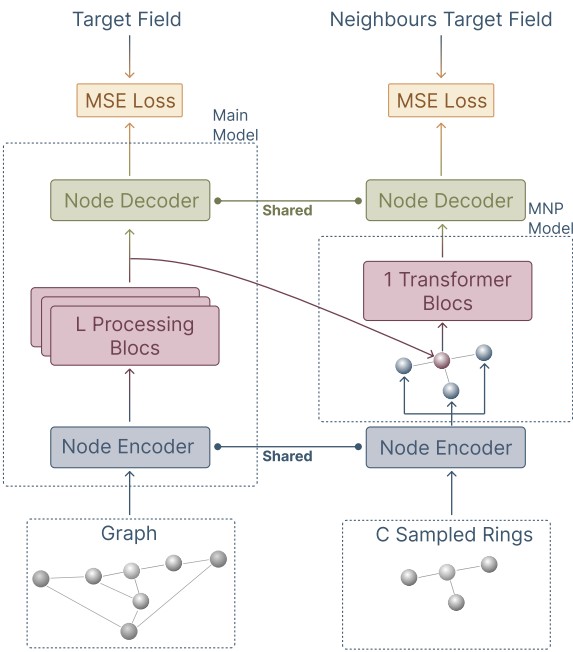

*Figure 1.* **Multi Node Prediction** We process each node with a given model. Before decoding each node, we construct rings that consist of a latent node and freshly encoded neighbors. We then train a small cross-attention layer to predict the fields of each neighbor, while most relevant information lives in the latent central node.

which does not explicitly enforce that the latent representation of $\mathbf{z}_\mathbf{i}^\mathbf{L}$ carries any information about the neighborhood (or stencil) around $i$. On irregular meshes with long-range, anisotropic dynamics, this omission manifests in several potential pathologies: (i) a lack of relevant information related to flux reconstruction in FEM; (ii) oversquashing, since relevant neighbor information must be compressed through message passing instead of living inside each node directly; (iii) stability issues, where small inconsistencies on a stencil around $i$ can amplify over rollouts, instead of having the ability to check itself against its neighbors.

To that end, we make a small but effective modification to our supervised task: Multi-Node Prediction. Let $C \subset V$ be a set of centers sampled per step and $\mathcal{N}(i)$ the 1-hop neighborhood of $i$. Each center is sampled uniformly from internal nodes only; boundary nodes are excluded as centers, the number of centers is fixed per batch, and centers are resampled at each training step. We form a *star sequence*:

$$\mathbf{S_i} = \begin{bmatrix} \mathbf{z_i^L} & \mathbf{z_{j_1}^0} & \dots & \mathbf{z_{j_{|\mathcal{N}(i)|}}^0} \end{bmatrix} \quad (8)$$

where $\mathbf{z_{j_k}^0} = \mathcal{E}(\mathbf{x_{j_k}^0})$ is the encoded feature of node $j_k$, a neighbor of node $i$, represented by its latest latent version $\mathbf{z_i^L}$. The goal is to force the latent representation of node $i$ to predict its own next step value, **but also** the one of

its neighbors, only from their encoded values and its own final latent representation. We define a small one-layer transformer $T_{\theta'}$ using ring self-attention (neighbors attend to the center and to each other, but stars are isolated and do not attend to themself) to yield:

$$\mathbf{O_i} = T_{\theta'}(\mathbf{S_i}) \in \mathbb{R}^{(|\mathcal{N}(i)|+1)\times d} \quad (9)$$

We then compute the same supervised loss as before between each decoded neighbor (using the same decoder $\mathcal{D}$), and its target field:

$$\mathcal{L}_{\text{MNP}}(\theta, \theta') = \frac{1}{|C|}\sum_{i\in C}\frac{1}{|\mathcal{N}(i)|}\sum_{j\in\mathcal{N}(i)}\ell(\mathcal{D}(\mathbf{O_{ij}}), y_j) \quad (10)$$

and define the total objective $\mathcal{L} = \mathcal{L}_{\text{main}} + \alpha\,\mathcal{L}_{\text{MNP}}$ with $\alpha = 0.2$. The overall architecture is presented in Figure 1. Overall, we shift $\mathbf{z_i^L}$ from being only predictive of $y_i$ to being jointly predictive of $\{y_i, \{y_j\}_{j\in\mathcal{N}(i)}\}$. More precisely, in Theorem B.2, we prove that patch reconstruction accuracy controls a discrete gradient (hence flux) error.

$$\frac{1}{N}\sum_{i=1}^{N}\left\|\widehat{\nabla}_h u(\mathbf{x}_i) - \nabla u(\mathbf{x}_i)\right\|^2 \leq \frac{C}{h^2}\Big(\mathcal{L}_{\text{MNP}} + \mathcal{L}_{\text{node}}\Big) + $$
$$C\,h^2\,\|u\|^2_{C^2(\Omega)} \quad (11)$$

This proves that $\mathcal{L}_{\text{MNP}}$ is a principled proxy for a Sobolev-type regularization that aligns the learned representation with the PDE's spatial operator.

---

**Algorithm 1** Multi-Node Prediction

1: Sample centers $C \subseteq V_{\text{int}}$ uniformly from internal nodes with $|C| = m$
2: **for all** $i \in C$ **do**
3:     $\mathcal{N}(i) \leftarrow \{j \in V : (i, j) \in E\}$         $\triangleright$ 1-hop
4:     Order $\mathcal{N}(i)$ as $(j_1, \dots, j_{K_i})$ where $K_i = |\mathcal{N}(i)|$
5:     $S_i \leftarrow [\mathbf{z}_i^L; \mathbf{z}_{j_1}^0; \dots; \mathbf{z}_{j_{K_i}}^0]$      $\triangleright$ star sequence
6: **end for**
7: $O_i \leftarrow T_{\theta'}(S_i)$   $\triangleright$ use a block-diagonal attention mask so that different stars do not attend
8: **for all** $i \in C$ **do**
9:     **for** $k = 1$ **to** $K_i$ **do**
10:         $j \leftarrow j_k$
11:         $\hat{y}_{j|i} \leftarrow \mathcal{D}(O_{i,k+1})$ $\triangleright$ token $k{+}1$ corresponds to neighbor $j_k$ (token 1 is the center)
12:         $\mathcal{L}_{\text{MNP}} \leftarrow \mathcal{L}_{\text{MNP}} + \frac{1}{|C|K_i}\ell(\hat{y}_{j|i}, y_j)$
13:     **end for**
14: **end for**

---

The computational overhead is only minor, since we simply add a single cross-attention layer and re-use the encoder $\mathcal{E}$ and decoder $\mathcal{D}$. Having hooks within the training framework also allows for most operations to be run only once. To keep our addition lightweight even on very large meshes, we pack the star sequences in a vectorized fashion with padding, capping the number of neighbors per center to $K$. Our additional module is used only during training, resulting in a 5% increase in training time. During our ablation study, we only examine the impact of this new task on performance, the effect of the number of centers we randomly select at each step (including the increase in training time), and the impact of this task on the nodes' latent representations. We add this method in our ablation study, and both study the impact of the number of centers chosen and other supervised approaches, such as an $L_2$ loss on the field's gradient.

## 3.2. Temporal Correction

Recall that many discrete-time machine learning surrogates generate next-step predictions with the operation $\mathbf{u}_{t+1} = \mathbf{u}_t + \Phi(\mathbf{u}_t)$, where $\Phi$ is made of $L$ spatial processing layers: $\mathbf{Z}^{\ell+1} = \mathbf{Z}^\ell + \phi_\ell(\mathbf{Z}^\ell)$. We argue that one should actually see the $L$ spatial updates as $L$ predictor-corrector (Gragg & Stetter, 1964) layers, *i.e.* as $L$ discretizations of the interval $[t, t+\Delta t]$, with the following formulation:

$$
\begin{aligned}
\tilde{\mathbf{Z}}^{\ell+1} &= \mathbf{Z}^\ell + \phi_\ell^s(\mathbf{Z}^\ell) \\
\mathbf{Z}^{\ell+1} &= \mathbf{Z}^\ell + \phi_\ell^t(\tilde{\mathbf{Z}}^{\ell+1}, \mathbf{Z}^\ell)
\end{aligned}
\tag{12}
$$

where $\phi_\ell^s$ is the same (spatial) processor as defined before, specific to each architecture, and $\phi_\ell^t$ is a learnable temporal corrector. Importantly, we do not increase the temporal context of our model; we still use a single timestep to predict the next. However, we interpret each processor block $\ell$ as a predictor in $[t + \Delta t \frac{\ell}{L}, t + \Delta t \frac{\ell+1}{L}]$. Equation (12) separates space from time while keeping a residual identity path that preserves gradients and reduces drift across rollouts.

Anisotropic PDEs require directionally selective stencils. While standard approaches are often largely isotropic, we implement our temporal corrector using cross-attention with queries from the predicted state, keys and values from the previous state, thereby routing updates along directions encoded by the evolving field itself. To overcome classical GNN pathologies such as oversquashing and oversmoothing, we also introduce a gating mechanism in the temporal corrector.

Let $\mathbf{C} = [\tilde{\mathbf{Z}}^{\ell+1}, \mathbf{Z}^\ell]$. Let $G([\tilde{\mathbf{Z}}, \mathbf{Z}]) \in [0, 1]^N$ be a gating mechanism implemented as a 2-layer MLP with a softmax for the final activation function. Let $M([\mathbf{h}^{\ell+1}, \mathbf{h}^\ell])$ be a mixing MLP with 2 layers. We implement the temporal corrector as follows:

| | |
|---|---|
| **Spatial** | $\tilde{\mathbf{Z}}^{\ell+1} = \mathbf{Z}^\ell + \phi_\ell^s(\mathbf{Z}^\ell)$ |
| **Temporal Cross Att.** | $\bar{\mathbf{Z}}^{\ell+1} = G(\mathbf{C}) \odot \mathrm{CA}(\mathbf{C})$ 
 $\phi_\ell^t(\mathbf{C}) = \bar{\mathbf{Z}}^{\ell+1} + M(\mathbf{C})$ |
| **Correction** | $\mathbf{Z}^{\ell+1} = \mathbf{Z}^\ell + \phi_\ell^t(\mathbf{C})$ |

$$\tag{13}$$

where CA is cross-attention and $\odot$ is the Hadamard product. During our ablation study, we investigate both the impact of the gating and mixing mechanism, and the trade-off between the frequency of said temporal correction and the increase in training time. In Theorem B.3, we show that the temporal correction block enlarges the class of stable one-step maps that the network can realize compared to an explicit residual update, thus allowing it to realize a larger class of stable one-step maps under the theorem assumptions than the standard approach. We add a temporal corrector in our ablation study, and investigate frequently to apply it and the impact of the gating and mixing mechanism.

## 3.3. 3D RoPE

Irregular meshes exhibit strong, orientation-dependent transport (e.g., advection along streamlines, anisotropic diffusion along fiber directions). Message-passing baselines and geometry-agnostic attention tend to learn isotropic mixing, hurting long-range fidelity. While using relative position in edges, or absolute position in nodes, can mitigate those issues, we argue that they should work in relation to another sort of positional encoding.

Injecting 2D/3D rotary structure into queries Q and keys K makes attention explicitly sensitive to relative 2D/3D offsets, which improves directional selectivity and long-horizon accuracy in our rollouts while preserving efficiency for neighbor-only graphs.

RoPE applies an axis-wise orthogonal rotation to Q and K with angles proportional to position. For one 2D channel pair $(u, v)$ and scalar angle $\theta$, $\mathrm{Rot}_\theta[u, v] = [u\cos\theta - v\sin\theta,\ u\sin\theta + v\cos\theta]$. If $R(\theta_i)$ and $R(\theta_j)$ are rotations applied to $q_i$ and $k_j$, then

$$
\langle R(\theta_i)q,\ R(\theta_j)k \rangle = \langle q,\ R(\theta_j - \theta_i)k \rangle
$$

so the score depends on *relative* angle. Extending to 3D, we assign disjoint channel pairs to the $x$, $y$, and $z$ axes and rotate with angles $\theta_{i,r}^{(a)} = \omega_r \tilde{p}_{i,a}$, where $\tilde{p}_i = p_i - \bar{p}$ are centered coordinates and $\{\omega_r\}$ are multi-scale frequencies. The dot-product remains a function of per-axis *relative* offsets $(\tilde{p}_j - \tilde{p}_i)$, which (i) encodes oriented distances that match anisotropic PDE fluxes and (ii) is translation-invariant by construction.

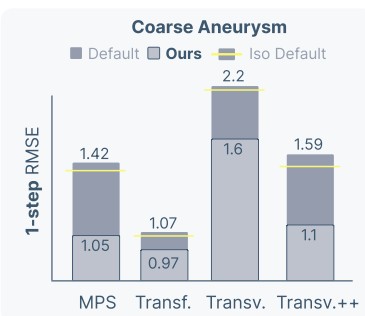 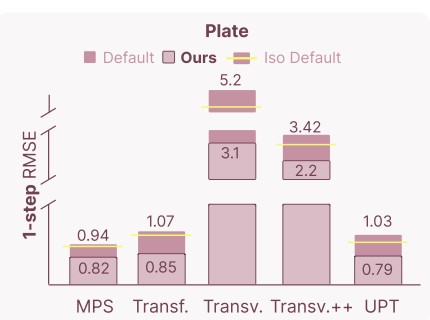 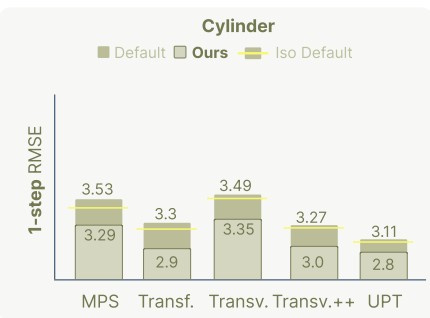

*Figure 2.* **Results on 1-step RMSE.** We see improvements for all models and all datasets. Since our approach incurs a slight increase in the number of trainable parameters, we also compute the metric for the original architecture with the same number of parameters as ours (represented by the yellow lines).

For each head of width $d_h$, we allocate channel pairs across the three axes. Let $\mathcal{I}_x, \mathcal{I}_y, \mathcal{I}_z$ index those pairs. With centered coordinates $\tilde{p}_i = (\tilde{x}_i, \tilde{y}_i, \tilde{z}_i)$ and frequencies $\{\omega_r\}$, we define:

$$\forall (2r, 2r+1) \in \mathcal{I}_x :$$
$$\begin{bmatrix} \tilde{q}_{i,2r} \\ \tilde{q}_{i,2r+1} \end{bmatrix} = \begin{bmatrix} \cos(\omega_r \tilde{x}_i) & -\sin(\omega_r \tilde{x}_i) \\ \sin(\omega_r \tilde{x}_i) & \cos(\omega_r \tilde{x}_i) \end{bmatrix} \begin{bmatrix} q_{i,2r} \\ q_{i,2r+1} \end{bmatrix}$$

and similarly for $\mathcal{I}_y, \mathcal{I}_z$, and the keys $\tilde{k}_j$

We then compute attention using $(\tilde{q}_i, \tilde{k}_j)$. This preserves neighbor sparsity, adds no parameters, and composes cleanly with all attention-based operations.

During our ablation study, we investigate whether RoPE helps with the performance or not. We also compare it to several other positional encoding approaches: a learnable positional embedding, a learnable relative position bias, and weighting the adjacency matrix by nodes' Euclidean distances.

## 4. Experiments

**Datasets**   We evaluated our models across different use cases. The first dataset is the flow past a cylinder (Pfaff et al., 2021) simulated using the COMSOL solver. The second dataset is the flow inside a brain aneurysm (Goetz et al., 2024), simulated using the CIMLIB (Digonnet et al., 2007) solver. Finally, the last dataset is the Deforming-Plate (Pfaff et al., 2021). Details such as the attributes used and the simulation time step $\Delta t$ are shown in Table 1 and Figure 7. Cylinder and DeformingPlate use the full MeshGraphNet configuration with 1000 training trajectories and 100 test trajectories; Coarse Aneurysm uses 100 training trajectories and 20 test trajectories. The trajectories contain 600, 400, and 80 steps for Cylinder, Deforming-Plate, and Aneurysm, respectively; $\Delta t = 0.01$s for Cylinder and Aneurysm. While the Cylinder and DeformingPlate

datasets consist of relatively small graphs (between 1 and 2,000 nodes), the brain aneurysm meshes are one order of magnitude larger. Following MeshGraphNet, each node is assigned a node type. During prediction, velocity boundary values are enforced for all node types except Normal and Outflow, while pressure is enforced only at Inlet nodes; MNP centers are sampled from internal nodes, although boundary nodes may still appear as neighbors and are handled with the same boundary-value enforcement rules.

**Training procedures**   To evaluate our models, we use the 1-step RMSE and the All-Rollout RMSE defined in (Sanchez-Gonzalez et al., 2020). Unless stated otherwise, RMSE values are averaged over predicted variables, nodes, timesteps, trajectories, and five random seeds; lower is better, and standard deviations are reported in ablation plots when space permits. Models were trained for 20 epochs with an AdamW (Loshchilov & Hutter, 2019) optimizer using $\beta_1 = 0.9, \beta_2 = 0.95$. We used a learning rate schedule with a warm-up period of 1000 steps, cosine decay, and a maximal learning rate of $10^{-3}$. Each model is trained on 5 different seeds. We reproduce the standard deviation obtained from the ablations. We introduce noise to our inputs using the same strategy as (Sanchez-Gonzalez et al., 2020). More specifically, we add random noise $\mathcal{N}(0, \sigma)$ to dynamical variables (see Table 2) at each training step and train solely on next-step prediction.

## 5. Results

Across the three different datasets and the three different models, we obtain improvements on all rollout metrics between 20 and 30 percent, for only a 10% increase in both training or inference time, using the optimal configurations defined in the ablation. (See Figure 2 for 1-step improvements and Figure 8 for more general results). We want to emphasize that our approach improves results across highly diverse datasets (2D, 3D, fluids, solids, varying numbers of

nodes) and models (message-passing-based GNN, attention-based GNN, attention-based point clouds). Importantly, models that match the total number of parameters in our architecture by increasing their width obtain only minor improvements. We believe this makes our approach a highly efficient method for improving the performance of mesh-based simulation, regardless of the dataset or architecture.

Finally, we tested geometry-shift generalization by training on a single-cylinder configuration and evaluating on multiple cylinders and on different shapes. These supplementary rollouts show that the proposed modules remain beneficial under unseen geometries and new vortex/recirculation patterns, but this test should be interpreted as a geometry-shift evaluation rather than a broad Reynolds-number generalization study.

We also study whether our approach scales with training time and the model's size, or if those improvements decrease as we scale up. In Figure 3, we present the results of a Transformer architecture on each dataset, as we scale the parameters from 500k (S), to 3M (M), 14M (L), and 51M (XL). For the Aneurysm and the Deforming Plate dataset, our approach scales with the number of parameters, yielding consistent results. On the Cylinder dataset, while our approach still yields better results, the improvements are less consistent. This may be because of benchmark saturation, as noted in (Garnier et al., 2025b).

In Figure 4, we study a message passing GNN on the Aneurysm and the Cylinder dataset. We scale the training times from 5 to 30 epochs for the Aneurysm dataset, and from 5 to 15 for the Cylinder dataset. Similarly to the previous experiment, we find that our approach robustly scales with training time, yielding consistent improvements across different numbers of gradient descent steps.

### 5.1. Ablation Studies

Our ablation study was run on the Aneurysm dataset, where each graph has, on average, 20k nodes, and we used Transformer models.

**Multi Node Prediction**   We study the impact of our Multi Node Prediction task, and the impact of the number of centers we use. Overall, we find that using as few as 256 centers (around 2.5% of the nodes) per trajectory yields improvements on both 1-step and all-rollout RMSE. We also find that improvements scale with the number of centers selected. Using 1024 centers gives twice the improvements from 256 centers. Results are presented in Figure 5-middle column. We also demonstrate in Figure 10 that our method offers much better results than replacing $\mathcal{L}_{\mathrm{MNP}}$ by a supervised loss on the field's gradient $\ell(\nabla \hat{y}_i, \nabla y_i)$. We also tested MNP centers sampled uniformly, biased toward the DeformingPlate contact/constraint region, and biased away from it;

the three configurations gave similar improvements, with no statistically significant difference from uniform internal-node sampling.

**Temporal Correction**   We find that our temporal correction yields strong improvement over 1-step and all-rollout RMSE. We first study the impact of our gating and mixing mechanism. Only using cross-attention worsens the performance, to a higher degree than using no temporal correction. This highlights the importance of those two architectures. We also study the impact of the number of temporal corrections. We compare using a correction term after each spatial block, or only after the last one. Both improve the performance, and there's a strong trade-off to find between computation time and better performance. However, those extra parameters are better consumed here than in extra spatial computation, as shown in our previous section. Finally, we also tried to add an extra loss term after each temporal correction. That is, after each intermediate $\ell$ step, we decoded the latent graph and used it as an intermediate supervised signal. This doesn't work well at all. Results are presented in Figure 5-right column.

**3D RoPE**   Overall, 3D RoPE provides the same performance as using absolute coordinates in the node feature. However, using both at the same time yields strong improvements. On the other hand, any other tested solution provides worse performance. We studied a learned positional embedding, that is, a learnable mapping between absolute position and $\mathbb{R}^d$, which we could sum with the node features. A learnable relative bias, that is, a learnable mapping between differences in absolute position and $\mathbb{R}^d$, we could sum after computing $Q^t K$. Finally, we also tried for the Transformer architecture to replace the unitary adjacency matrix with a weighted version. None of these three approaches yields improvements. Results are presented in Figure 5-left column.

The rest of the ablation are presented in Appendix C.

### 5.2. Other Experiments

**Latent Representation**   To study the impact of Multi-Node Prediction on Latent Representation, we conduct a simple experiment. During training, after each step of a model, we compute its $L_2$ difference with the latent target (by latent target, we mean the encoded version of the next-step target). In Figure 11, we present the results on a message-passing GNN with and without Multi-Node Prediction. We obtain similar results on all datasets and all architectures. First, we can see that this approach offers a latent representation that is much closer to the target. Second, while not optimizing it at all, we can see that this loss remains stable or even decreases, while it increases without Multi-Node Prediction. We believe this is a key explanation as to why this new supervised task offers such improvements

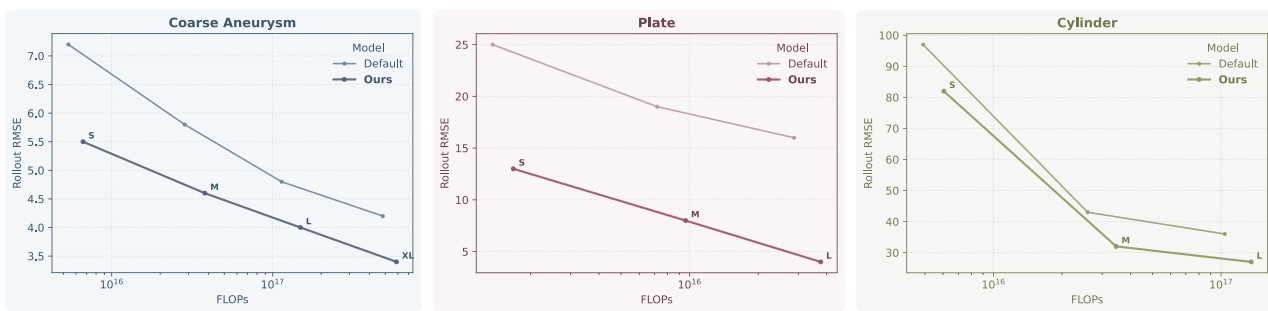

*Figure 3.* **Scaling with model size.** All rollout RMSE for a Transformer model on the three different datasets. Even when training models larger and larger, our approach keeps scaling similarly to the previous model.

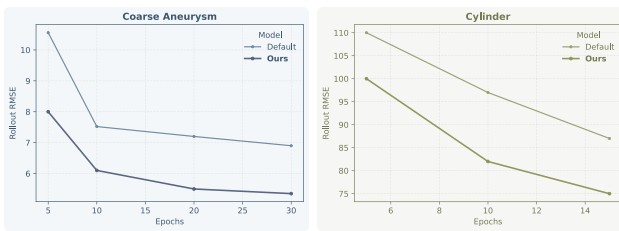

*Figure 4.* **Scaling with training time** All rollout RMSE for a MeshGraphNet architecture of two different datasets, for different training schedules.

on long-term rollout.

**Predicting subtasks**   We used the latent representation of each layer (layer 0 being post-Encoder) of a Transformer with 10 layers, with and without Multi-Node Prediction training. We then fit a simple 2-layer MLP to predict different quantities such as the next step Velocity, Pressure and Wall Shear Stress (WSS) Figure 6. It's important to note that while the current step velocity is present in the inputs, that is not the case for the pressure and the WSS. We first find that the latent representation obtained using Multi-Node Prediction offers lower loss. This will be constant throughout all experiments. While predicting the pressure, similarly to some tasks in computer vision, we find that the errors decrease before increasing again, a sweet spot being around 2/3 of the spatial computation. Finally, since it is extremely related to the velocity field and its differential outcomes, WSS prediction decreases with the amount of spatial processing. Interestingly, a model trained without Multi-Node Prediction doesn't see such a decrease.

**Aneurysm physical metrics**   Because RMSE alone does not guarantee physical validity, we also evaluated spatially averaged WSS over all timesteps on three Aneurysm test cases and mass flow rate over time at four 2D planes inside the main artery. These supplementary plots show stable physical metrics throughout autoregressive rollout, even though the final model is not trained with a physics-informed

loss.

## 6. Conclusion

In this study, we presented 3 improvements for standard approaches to mesh-based simulations. We demonstrated that those approaches increase performance on both short-term and long-term prediction, while improving the latent representation of the graph throughout training. Importantly, our methods are architecture and dataset-agnostic across the tested settings.

## Acknowledgements

The authors acknowledge the financial support from ERC grant no 2021-CoG-101045042, CURE. Views and opinions expressed are however those of the author(s) only and do not necessarily reflect those of the European Union or the European Research Council. Neither the European Union nor the granting authority can be held responsible for them.

The authors thank Arthur Verrez and Loïc Chadoutaud for valuable feedback on the manuscript.

## Impact Statement

This paper presents work whose goal is to advance the field of Machine Learning. There are many potential societal consequences of our work, none which we feel must be specifically highlighted here.

## Code availability

Code, configuration files, and evaluation scripts and datasets are available at `https://github.com/DonsetPG/graph-physics`.

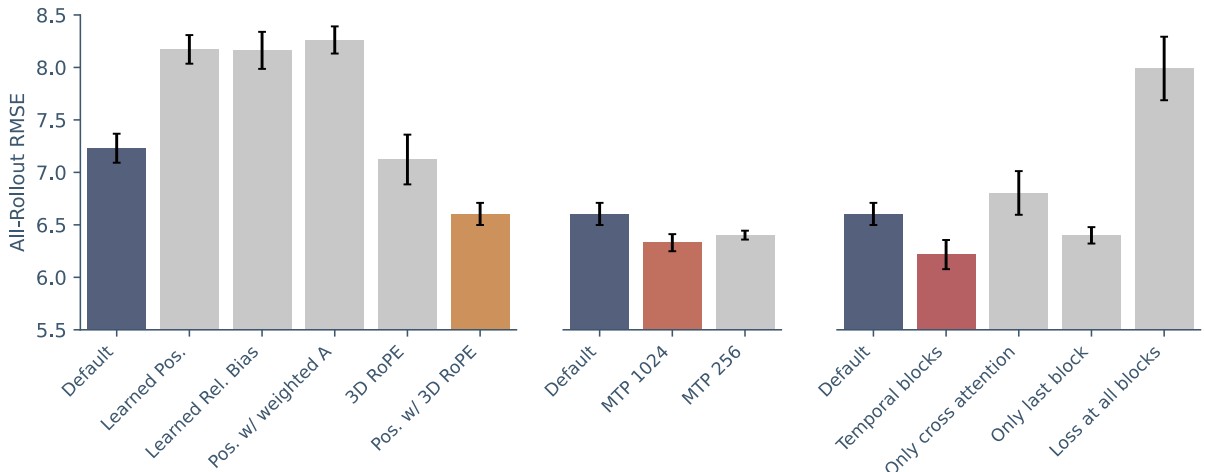

*Figure 5.* **Ablation Studies** We present our ablation studies for all three improvements and their alternatives. Ablations related to RoPE are in the left, to Multi-node Prediction in the middle, and to Temporal Correction in the right.

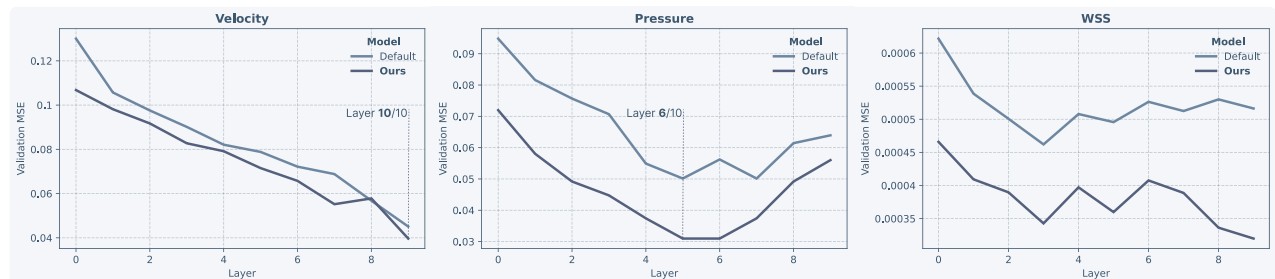

*Figure 6.* **Subtasks prediction.** We compute a regression task on next-step fields: velocity, pressure, and WSS, using latent representations from different layers. Importantly, the *Default* architecture is not always the same per ablation. For example, the default architecture for Multi Node Prediction is an architecture using RoPE.

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

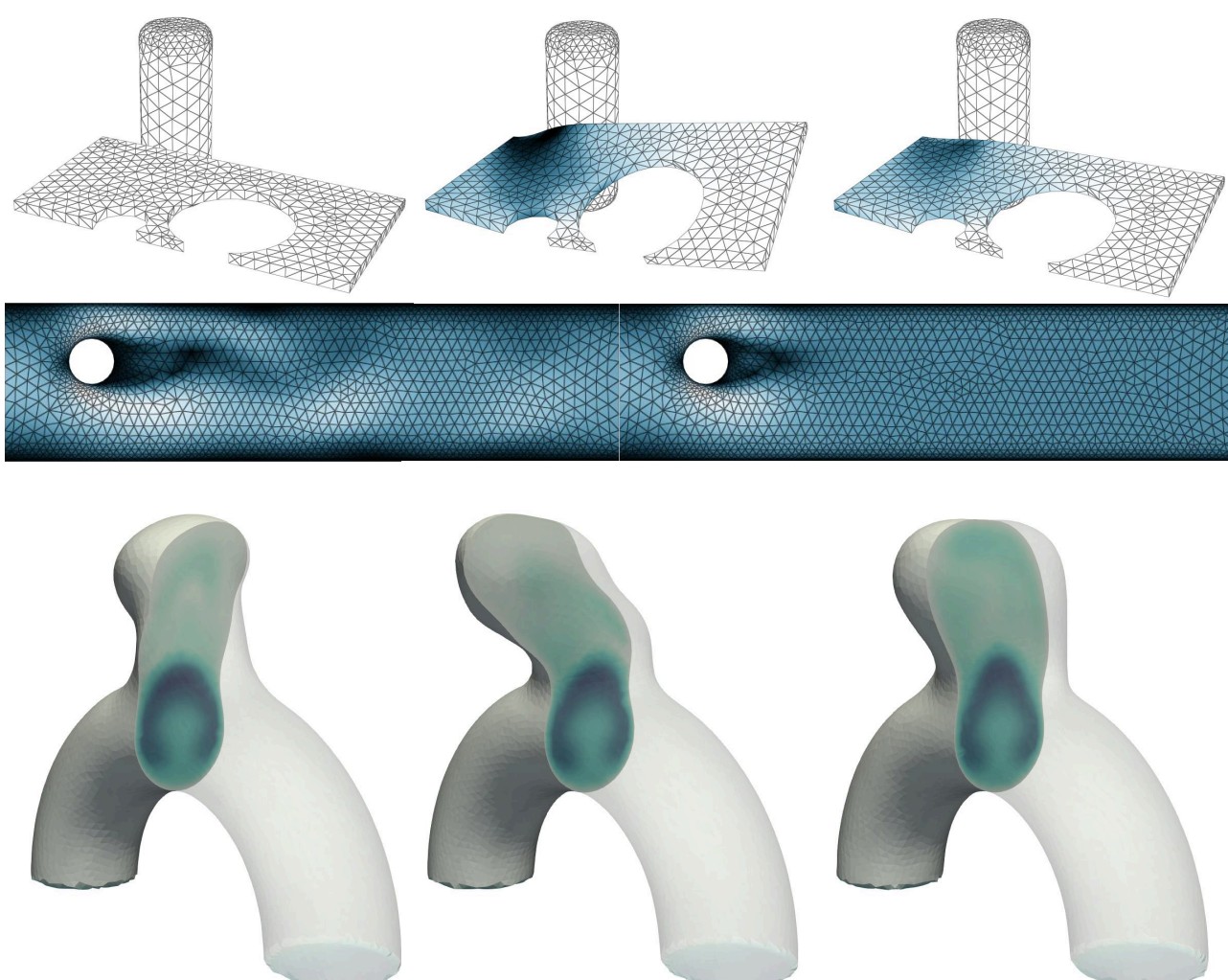

*Figure 7.* We display details of our three primary datasets: DeformingPlate, Cylinder and Coarse Aneurysm. For the first two, we also display the mesh used for the simulations, while we discard it from the Aneurysm visualisation for practicality purposes.

## A. Datasets

We give details below about the inputs and outputs used for each dataset (see Table 1 and Figure 7). Cylinder and DeformingPlate were generated with COMSOL (multiphysics®, 2020) and were introduced by (Pfaff et al., 2021). Aneurysm was generated with CimLib (Digonnet et al., 2007) and was introduced by (Goetz et al., 2024; Garnier et al., 2025a).

In Table 1, $n$ is the node type (Inflow, Outflow, Wall, Obstacle, Normal) and $v_{\text{in}}$ the inflow velocity at the current timestep. Boundary values are enforced at prediction time using these node types: velocity is enforced for all node types except Normal and Outflow, and pressure is enforced only for Inlet nodes. When history is different than 0, we use a first-order derivative of the inputs as an extra feature. For example, we add $a_x, a_y, a_z$ to each node from an aneurysm mesh.

### A.1. Noise

We display the noise used as well as the global nodes selected for each dataset in Table 2.

**Noise Scale**  We make our inputs noisy by following the same strategy as (Sanchez-Gonzalez et al., 2020). We add random noise $\mathcal{N}(0, \sigma)$ to the dynamical inputs. Each noise standard deviation was selected using the following procedure. We train one model without any noise and then compute the distribution of the one-step error. We use this error deviation as

*Table 1.* Datasets used in the experiments, their features as well as if some temporal history is used. Finally, we also present the average number of nodes per mesh, together with solver, train/test trajectories, rollout length, and time step.

| Dataset | Inputs | Outputs | History | Nodes | Solver | Train | Test | Steps | $\Delta t$ |
|---|---|---|---|---|---|---|---|---|---|
| Cylinder | $n, v_x, v_y$ | $v_x, v_y$ | 0 | 2000 | COMSOL | 1000 | 100 | 600 | 0.01s |
| DeformingPlate | $n, x, y, z, f_{\text{in}}$ | $x, y, z, \sigma$ | 0 | 1000 | COMSOL | 1000 | 100 | 400 | − |
| Coarse Aneurysm | $n, v_x, v_y, v_z, v_{\text{in}}$ | $v_x, v_y, v_z$ | 1 | 20000 | CimLib | 100 | 20 | 80 | 0.01s |
| Fine Aneurysm | $n, v_x, v_y, v_z, v_{\text{in}}$ | $v_x, v_y, v_z$ | 1 | 250000 | CimLib | 100 | 20 | 80 | 0.01s |

*Table 2.* Noise levels applied to each dataset, per feature, during training.

| Dataset | Noise |
|---|---|
| Cylinder | 0.02 |
| Plate | 0.003 |
| 3D-Aneurysm | $v_x, v_y$: 10, $v_z$: 1 |

the noise scale. For the case of the aneurysm dataset, the variational nature of the inflow makes for said distribution to be time-dependant. While we investigated different strategies, such as time-dependant noise, we simply used the larger standard deviation possible($\max\limits_{t \in [1, T-1]} \sigma_t$).

# B. Theoretical Analysis

We provide theoretical grounding for our architectural contributions by connecting them to two classical themes in numerical PDEs: (i) *spatial consistency*, where stability and accuracy depend on controlling derivative (flux) errors rather than only pointwise errors; and (ii) *time-integration stability*, where explicit one-step methods are prone to drift for stiff semi-discrete systems. Throughout, we view the mesh/graph as nodes $\{\mathbf{x}_i\}_{i=1}^N \subset \Omega \subset \mathbb{R}^d$ with edges given by mesh adjacency. We denote the (maximum) local mesh size by $h := \max_{(i,j) \in E} \|\mathbf{x}_j - \mathbf{x}_i\|$ and the neighborhood of node $i$ by $\mathcal{N}(i)$. For simplicity the analysis is stated for a scalar field $u : \Omega \to \mathbb{R}$; the vector-valued case $u : \Omega \to \mathbb{R}^c$ follows componentwise with identical bounds.

## B.1. Multi Node Prediction as Sobolev Regularization

Standard surrogates usually minimize a node-wise $L^2$ error:

$$\mathcal{L}_{\text{node}} := \frac{1}{N} \sum_{i=1}^N \left\| u(\mathbf{x}_i) - \hat{u}_i \right\|^2$$

which controls pointwise accuracy but does not directly control errors in spatial derivatives. Yet, many PDE operators (in strong or weak form) depend on gradients and fluxes, e.g. diffusion energies $\int_\Omega \|\nabla u\|^2$ or nonlinear fluxes $f(u, \nabla u)$. Consequently, purely node-wise supervision can allow locally inconsistent stencils (large derivative/flux errors) even when pointwise errors are small, a behavior that is particularly costly in long rollouts. MNP augments node-wise supervision with *patch* supervision: for each center node $i$, the model produces predictions for its neighborhood:

$$\hat{u}_{j|i} \approx u(\mathbf{x}_j) \qquad j \in \mathcal{N}(i)$$

In our implementation (Sec. 3.1), these predictions are produced by a one-layer star transformer that processes the center latent together with freshly encoded neighbor features. Formally, one can write:

$$(\mathbf{o}_{k|i})_{k \in \{i\} \cup \mathcal{N}(i)} = T_{\theta'}\Big([\mathbf{z}_i^L; \{\mathbf{z}_j^0\}_{j \in \mathcal{N}(i)}]\Big) \qquad \hat{u}_{j|i} = \mathcal{D}_\theta(\mathbf{o}_{j|i})$$

but the analysis below is *architecture-agnostic*: it only depends on the resulting scalar predictions $\hat{u}_{j|i}$ and therefore applies equally to (i) direct decoding from $\mathbf{z}_i^L$ and (ii) the implemented auxiliary-transformer variant.

We define the MNP loss as the following:

$$\mathcal{L}_{\text{MNP}} := \frac{1}{N} \sum_{i=1}^{N} \sum_{j \in \mathcal{N}(i)} w_{ij} \left\| \hat{u}_{j|i} - u(\mathbf{x}_j) \right\|^2 \tag{14}$$

for nonnegative weights $w_{ij}$. Importantly, the discrete gradients used below depend on differences $(v_j - v_i)$ and therefore necessarily involve the *center* error $|\hat{u}_i - u(\mathbf{x}_i)|$. In our training setup this term is provided by $\mathcal{L}_{\text{node}}$ (i.e., $\mathcal{L}_{\text{main}}$ in Sec. 3.1).

Given values $\{v_j\}_{j \in \{i\} \cup \mathcal{N}(i)}$, define the *local weighted least-squares* (WLS) discrete gradient $\nabla_h v(\mathbf{x}_i) \in \mathbb{R}^d$ as:

$$\nabla_h v(\mathbf{x}_i) := \arg\min_{\mathbf{g} \in \mathbb{R}^d} \sum_{j \in \mathcal{N}(i)} w_{ij} \left| (v_j - v_i) - \mathbf{g}^\top (\mathbf{x}_j - \mathbf{x}_i) \right|^2 \tag{15}$$

Let $\mathbf{B}_i \in \mathbb{R}^{|\mathcal{N}(i)| \times d}$ have rows $(\mathbf{x}_j - \mathbf{x}_i)^\top$ and $\mathbf{W}_i := \text{diag}(w_{ij})$. With $\mathbf{d}_i(v) := (v_j - v_i)_{j \in \mathcal{N}(i)}$, the minimizer has the closed form

$$\nabla_h v(\mathbf{x}_i) = \left( \mathbf{B}_i^\top \mathbf{W}_i \mathbf{B}_i \right)^{-1} \mathbf{B}_i^\top \mathbf{W}_i \, \mathbf{d}_i(v) \tag{16}$$

whenever $\mathbf{B}_i^\top \mathbf{W}_i \mathbf{B}_i$ is invertible.

**Assumption B.1.** For every node $i$, the weights satisfy $w_{ij} \geq 0$ and are normalized as $\sum_{j \in \mathcal{N}(i)} w_{ij} = 1$. Moreover, there exist constants $0 < c_0 \leq c_1 < \infty$, independent of $h$ and $i$, such that the weighted moment matrix obeys the uniform spectral bounds

$$c_0 \, h^2 \, \mathbf{I}_d \; \preceq \; \mathbf{B}_i^\top \mathbf{W}_i \mathbf{B}_i \; \preceq \; c_1 \, h^2 \, \mathbf{I}_d \tag{17}$$

These conditions hold for quasi-uniform/shape-regular meshes with bounded-degree neighborhoods and standard distance-based weights normalized to unit sum, provided each neighborhood spans $\mathbb{R}^d$ with a uniform angle condition.

**Theorem B.2** (**MNP + node-wise supervision controls a discrete $H^1$ seminorm**). *Assume $u \in C^2(\Omega)$ and Assumption B.1 holds. Define the* predicted patch *at node $i$ by setting $v_i = \hat{u}_i$ (the standard node-wise prediction) and $v_j = \hat{u}_{j|i}$ for $j \in \mathcal{N}(i)$, and let $\widehat{\nabla}_h u(\mathbf{x}_i) := \nabla_h v(\mathbf{x}_i)$ be the WLS gradient (15) computed from these predicted values. Then there exists a constant $C > 0$, independent of $h$, such that for every node $i$,*

$$\left\| \widehat{\nabla}_h u(\mathbf{x}_i) - \nabla u(\mathbf{x}_i) \right\|^2 \; \leq \; \frac{C}{h^2} \left( \sum_{j \in \mathcal{N}(i)} w_{ij} \left| \hat{u}_{j|i} - u(\mathbf{x}_j) \right|^2 + \left| \hat{u}_i - u(\mathbf{x}_i) \right|^2 \right) \; + \; C \, h^2 \, \|u\|_{C^2(\Omega)}^2 \tag{18}$$

*Consequently,*

$$\frac{1}{N} \sum_{i=1}^{N} \left\| \widehat{\nabla}_h u(\mathbf{x}_i) - \nabla u(\mathbf{x}_i) \right\|^2 \; \leq \; \frac{C}{h^2} \left( \mathcal{L}_{\text{MNP}} + \mathcal{L}_{\text{node}} \right) \; + \; C \, h^2 \, \|u\|_{C^2(\Omega)}^2 \tag{19}$$

*The same result holds componentwise for vector-valued fields $u : \Omega \to \mathbb{R}^c$.*

*Proof.* Fix a node $i$ and denote $\mathbf{d}_{ij} := \mathbf{x}_j - \mathbf{x}_i$. Let $v$ and $\tilde{v}$ be two sets of patch values on $\{i\} \cup \mathcal{N}(i)$. From (16), we have

$$\nabla_h v(\mathbf{x}_i) - \nabla_h \tilde{v}(\mathbf{x}_i) = \left( \mathbf{B}_i^\top \mathbf{W}_i \mathbf{B}_i \right)^{-1} \mathbf{B}_i^\top \mathbf{W}_i \left( \mathbf{d}_i(v) - \mathbf{d}_i(\tilde{v}) \right)$$

Taking norms and using $\|\mathbf{B}_i^\top \mathbf{W}_i \mathbf{q}\| \leq \|\mathbf{B}_i^\top \mathbf{W}_i^{1/2}\| \, \|\mathbf{q}\|_{\mathbf{W}_i}$ yields

$$\left\| \nabla_h v(\mathbf{x}_i) - \nabla_h \tilde{v}(\mathbf{x}_i) \right\| \leq \left\| \left( \mathbf{B}_i^\top \mathbf{W}_i \mathbf{B}_i \right)^{-1} \right\| \, \left\| \mathbf{B}_i^\top \mathbf{W}_i^{1/2} \right\| \, \left\| \mathbf{d}_i(v) - \mathbf{d}_i(\tilde{v}) \right\|_{\mathbf{W}_i}$$

By Assumption B.1, $\|(\mathbf{B}_i^\top \mathbf{W}_i \mathbf{B}_i)^{-1}\| \leq (c_0 h^2)^{-1}$ and $\|\mathbf{B}_i^\top \mathbf{W}_i^{1/2}\|^2 = \lambda_{\max}(\mathbf{B}_i^\top \mathbf{W}_i \mathbf{B}_i) \leq c_1 h^2$, so we ahve the following:

$$\left\| \nabla_h v(\mathbf{x}_i) - \nabla_h \tilde{v}(\mathbf{x}_i) \right\| \leq \frac{\sqrt{c_1}}{c_0} \frac{1}{h} \left\| \mathbf{d}_i(v) - \mathbf{d}_i(\tilde{v}) \right\|_{\mathbf{W}_i} \tag{20}$$

Now set $v_i = \hat{u}_i$, $v_j = \hat{u}_{j|i}$ for $j \in \mathcal{N}(i)$, and $\tilde{v}_k = u(\mathbf{x}_k)$ for all $k$. Define the errors $e_i := \hat{u}_i - u(\mathbf{x}_i)$ and $e_{j|i} := \hat{u}_{j|i} - u(\mathbf{x}_j)$. Then for each $j \in \mathcal{N}(i)$:

$$\big(\mathbf{d}_i(v) - \mathbf{d}_i(\tilde{v})\big)_j = (\hat{u}_{j|i} - \hat{u}_i) - (u(\mathbf{x}_j) - u(\mathbf{x}_i)) = e_{j|i} - e_i$$

hence

$$\left\|\mathbf{d}_i(v) - \mathbf{d}_i(\tilde{v})\right\|_{\mathbf{W}_i}^2 = \sum_{j \in \mathcal{N}(i)} w_{ij} |e_{j|i} - e_i|^2 \leq 2 \sum_{j \in \mathcal{N}(i)} w_{ij} |e_{j|i}|^2 + 2\Big(\sum_{j \in \mathcal{N}(i)} w_{ij}\Big) |e_i|^2$$

Using the normalization $\sum_j w_{ij} = 1$ in Assumption B.1 and combining with (20) gives us:

$$\left\|\nabla_h \hat{u}_{\cdot|i}(\mathbf{x}_i) - \nabla_h u(\mathbf{x}_i)\right\|^2 \leq \frac{C}{h^2}\left(\sum_{j \in \mathcal{N}(i)} w_{ij} |e_{j|i}|^2 + |e_i|^2\right) \tag{21}$$

for a constant $C$ independent of $h$.

It remains to relate $\nabla_h u(\mathbf{x}_i)$ to the true gradient $\nabla u(\mathbf{x}_i)$. By Taylor expansion, for each $j \in \mathcal{N}(i)$, we have:

$$u(\mathbf{x}_j) - u(\mathbf{x}_i) = \nabla u(\mathbf{x}_i)^\top \mathbf{d}_{ij} + r_{ij} \qquad |r_{ij}| \leq C_1 \|u\|_{C^2(\Omega)} \|\mathbf{d}_{ij}\|^2 \leq C_1 \|u\|_{C^2(\Omega)} h^2$$

Evaluating the WLS objective (15) at $\mathbf{g} = \nabla u(\mathbf{x}_i)$ yields residuals $r_{ij}$ and thus

$$J_i(\nabla u(\mathbf{x}_i); u) = \sum_{j \in \mathcal{N}(i)} w_{ij} |r_{ij}|^2 \leq C_2 h^4 \|u\|_{C^2(\Omega)}^2$$

where we used again $\sum_j w_{ij} = 1$. Since $\nabla_h u(\mathbf{x}_i)$ minimizes $J_i(\cdot; u)$ and $J_i(\nabla_h u(\mathbf{x}_i); u) \geq 0$, strong convexity of $J_i$ with Hessian $2\mathbf{B}_i^\top \mathbf{W}_i \mathbf{B}_i$ implies

$$c_0 h^2 \left\|\nabla_h u(\mathbf{x}_i) - \nabla u(\mathbf{x}_i)\right\|^2 \leq J_i(\nabla u(\mathbf{x}_i); u) \leq C_2 h^4 \|u\|_{C^2(\Omega)}^2$$

hence

$$\left\|\nabla_h u(\mathbf{x}_i) - \nabla u(\mathbf{x}_i)\right\|^2 \leq C_3 h^2 \|u\|_{C^2(\Omega)}^2 \tag{22}$$

Finally, by the triangle inequality and $(a + b)^2 \leq 2a^2 + 2b^2$, combining (21) and (22) yields (18). Averaging (18) over $i$ gives (19), with $\mathcal{L}_{\text{MNP}}$ and $\mathcal{L}_{\text{node}}$ as defined above. □

Theorem B.2 formalizes MNP as a Sobolev-type regularizer: patch reconstruction accuracy controls a discrete gradient (hence flux) error. Importantly, because the WLS gradient uses differences $(v_j - v_i)$, the gradient bound *must* depend on both neighbor errors ($\mathcal{L}_{\text{MNP}}$) and the center error ($\mathcal{L}_{\text{node}}$). In our training objective $\mathcal{L} = \mathcal{L}_{\text{main}} + \alpha \mathcal{L}_{\text{MNP}}$, the center term is exactly $\mathcal{L}_{\text{main}}$ (up to the choice of $\ell$), so controlling the total loss controls an averaged discrete $H^1$-type error. If one additionally supervises the center token inside the MNP star, the same analysis holds with $\mathcal{L}_{\text{node}}$ absorbed into the patch loss.

### B.2. Temporal Correction as Predictor-Corrector Integration

A common neural rollout uses a ResNet-style one-step update $\mathbf{Z}^{t+1} = \mathbf{Z}^t + \Phi(\mathbf{Z}^t)$. When $\Phi(\mathbf{Z}^t) \approx \Delta t \, F(\mathbf{Z}^t)$ for some latent vector field $F$, this is algebraically equivalent to *Forward Euler* on the ODE $\dot{\mathbf{Z}} = F(\mathbf{Z})$. Forward Euler is only conditionally stable; for stiff semi-discrete PDE systems this produces long-horizon drift.

We interpret our temporal block as a learned predictor-corrector:

$$\begin{aligned}
\tilde{\mathbf{Z}}^{t+1} &= \mathbf{Z}^t + \Phi^s(\mathbf{Z}^t) \\
\mathbf{Z}^{t+1} &= \mathbf{Z}^t + \Phi^t(\tilde{\mathbf{Z}}^{t+1}, \mathbf{Z}^t)
\end{aligned} \tag{23}$$

where $\Phi^t$ is implemented by cross-attention between the predicted future $\tilde{\mathbf{Z}}^{t+1}$ (query) and the history/current state $\mathbf{Z}^t$ (keys/values), followed by a bounded gate. This design gives the model enough information to approximate integrators that depend on both $t$ and $t+1$ states, i.e. implicit or semi-implicit schemes.

**Theorem B.3** (**Stability region expansion via an embedded $\theta$-method**). *Consider the scalar linear test equation $y'(t) = \lambda y(t)$ with $\Re(\lambda) \leq 0$. For $\theta \in [0, 1]$, the classical $\theta$-method is defined by the one-step update*

$$y_{n+1} = y_n + \Delta t\big((1 - \theta)\lambda y_n + \theta\lambda y_{n+1}\big) \qquad \Longleftrightarrow \qquad y_{n+1} = R_\theta(z)\, y_n, \;\; z := \Delta t\, \lambda \tag{24}$$

*with amplification factor*

$$R_\theta(z) = \frac{1 + (1 - \theta)z}{1 - \theta z} \tag{25}$$

*Then:*

1. *(A-stability) If $\theta \geq \frac{1}{2}$, then $|R_\theta(z)| \leq 1$ for all $z \in \mathbb{C}$ with $\Re(z) \leq 0$.*

2. *(Strict expansion over Forward Euler) Forward Euler is the special case $\theta = 0$ with $R_0(z) = 1 + z$, which is stable only for $z$ in the disk $|1 + z| \leq 1$. In contrast, any $\theta \geq \frac{1}{2}$ is stable for all $\Re(z) \leq 0$, i.e. removes the CFL-type restriction for the test problem.*

*Moreover, the two-stage update (23) can represent the family (24) by learning an effective $\theta$ through the bounded gate and by using the future-conditioned correction to approximate the implicit dependence on $y_{n+1}$. Hence, the temporal correction block enlarges the class of stable one-step maps that the network can realize compared to an explicit residual update.*

*Proof.* The equivalence in (24) is obtained by collecting the $y_{n+1}$ terms:

$$(1 - \theta z)\, y_{n+1} = (1 + (1 - \theta)z)\, y_n$$

which yields (25) provided $1 - \theta z \neq 0$.

We now prove A-stability for $\theta \geq \frac{1}{2}$. Let $z = a + ib$ with $a = \Re(z) \leq 0$. Then

$$|R_\theta(z)|^2 = \frac{|1 + (1 - \theta)z|^2}{|1 - \theta z|^2} = \frac{(1 + (1 - \theta)a)^2 + (1 - \theta)^2 b^2}{(1 - \theta a)^2 + \theta^2 b^2}$$

Thus, $|R_\theta(z)| \leq 1$ is equivalent to showing the denominator is at least the numerator:

$$(1 - \theta a)^2 + \theta^2 b^2 - \Big((1 + (1 - \theta)a)^2 + (1 - \theta)^2 b^2\Big)$$
$$= \Big[(1 - \theta a)^2 - (1 + (1 - \theta)a)^2\Big] + \Big[\theta^2 - (1 - \theta)^2\Big]b^2$$

Expanding the squares,

$$(1 - \theta a)^2 - (1 + (1 - \theta)a)^2 = (1 - 2\theta a + \theta^2 a^2) - (1 + 2(1 - \theta)a + (1 - \theta)^2 a^2) = -2a + (2\theta - 1)a^2$$

and also

$$\theta^2 - (1 - \theta)^2 = \theta^2 - (1 - 2\theta + \theta^2) = 2\theta - 1$$

Therefore,

$$(1 - \theta a)^2 + \theta^2 b^2 - \Big((1 + (1 - \theta)a)^2 + (1 - \theta)^2 b^2\Big) = -2a + (2\theta - 1)(a^2 + b^2) \tag{26}$$

If $a \leq 0$, then $-2a \geq 0$; if $\theta \geq \frac{1}{2}$, then $2\theta - 1 \geq 0$, and $a^2 + b^2 \geq 0$. Hence the right-hand side of (26) is nonnegative, proving the denominator dominates the numerator and thus $|R_\theta(z)| \leq 1$. This establishes A-stability for $\theta \geq \frac{1}{2}$.

Finally, for $\theta = 0$ we have $R_0(z) = 1 + z$, so stability requires $|1 + z| \leq 1$, i.e. the classical Forward Euler stability disk, which excludes large negative real $z$ and therefore imposes a step-size restriction for stiff modes. This proves the strict stability region expansion claimed in the theorem. $\square$

**Interpretation.** Theorem B.3 formalizes the numerical role of the temporal correction block: a two-stage update that can emulate a $\theta$-method (e.g. $\theta = \frac{1}{2}$ corresponding to Crank-Nicolson / trapezoidal rule) has a fundamentally larger stability region than Forward Euler. In our architecture, the cross-attention (future-conditioned correction) supplies the information pathway needed to approximate the implicit dependence on the next state, while the bounded gate provides a mechanism to adapt the effective $\theta$ locally, damping stiff/high-frequency error modes that otherwise destabilize long rollouts.

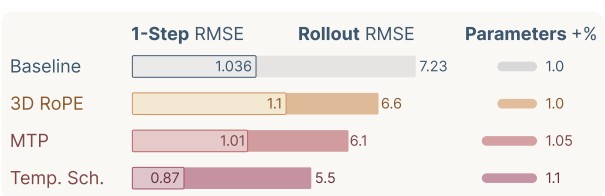

*Figure 8.* **Impact of each upgrade.** We detail the improvement in term of 1-step and all-rollout RMSE on the Aneurysm dataset for the Transformer architecture. We also detail the increase in terms of trainable parameters.

**Width vs. Depth** For a given budget (in terms of trainable parameters), we study the optimal number of layers to achieve the lowest all-rollout RMSE. We display the results of a Transolver architecture on the Cylinder Dataset. Overall, we find a clear bottleneck around 15 layers, no matter the increase in width. We find a similar number of optimal layers in other models, and in other datasets. We believe this is related both to the solver used to generate datasets, and to the graph based architecture that struggles with vanishing gradient after too many spatial processing (Di Giovanni et al., 2023). In lights of these results, we never go above 15 layers in any models we trained in this paper. Results are presented in Figure 9.

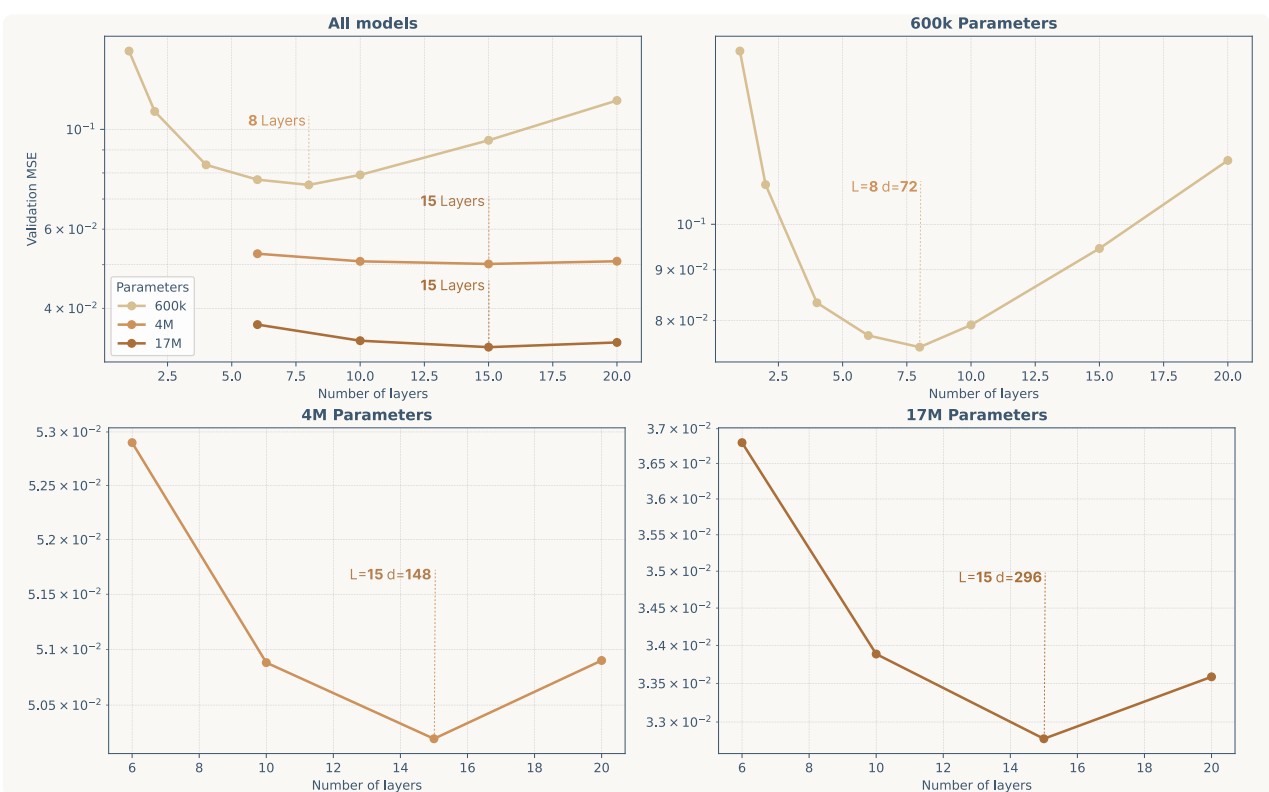

*Figure 9.* We study the performance of a transolver model under a fixed number of parameters. We train models with 600k, 4M and 17M parameters, and very accordingly the number of layers and the hidden dimension to keep the parameters constant.

## C. Additional experiments

**Additional Ablations** We present several additional experiments that were made in Figure 10. First, we investigate attention variations:

- Multi Head Latent Attention (MHLA) (Liu et al., 2024)
- Gated Delta Net in a 1 to 1 ratio (KimiTeam et al., 2025)
- Gated Attention (KimiTeam et al., 2025)

While MHLA does provide slightly better performance, it comes at a cost in terms of training time. Overall, we believe the tradeoff is not necessarily beneficial. We obtain similar findings to the two other approaches.

Finally, we also investigate with two additional losses. The first one consists of adding the divergence of the predicted velocity field, (which is supposed to be zero since the fluids are incompressible). This divergence residual improves 1-step RMSE, but it overfits the training distribution and increases all-rollout RMSE, so we do not retain it in the final framework. This observation is empirical for our setup and does not imply that physics-informed losses are generally ineffective. We also added a second supervised loss: the cosine similarity between the target and predicted fields. The goal of such a loss is to enforce the predicted fields to also align in terms of orientations with the target. This offers a small improvement in long term prediction.

**Latent Representation**    We also extract the latent representation after half the spatial processing with and without Multi Node Prediction, compute a PCA of said representation, and then plot the three main components as RGB pixels. Results are presented in Figure 12. We can see that the latent representation with MNP is much crisper and interesting that the one without MNP. More interestingly, the latent representation closely resemble the pressure field at this time, while it was never shown in any inputs to the model. The latent representation at the first and final layer does not exhibit such phenomenon, being much closely related to the velocity field itself. This suggest that models may actually be learning adjacent physics on top of the task they are supervised on.

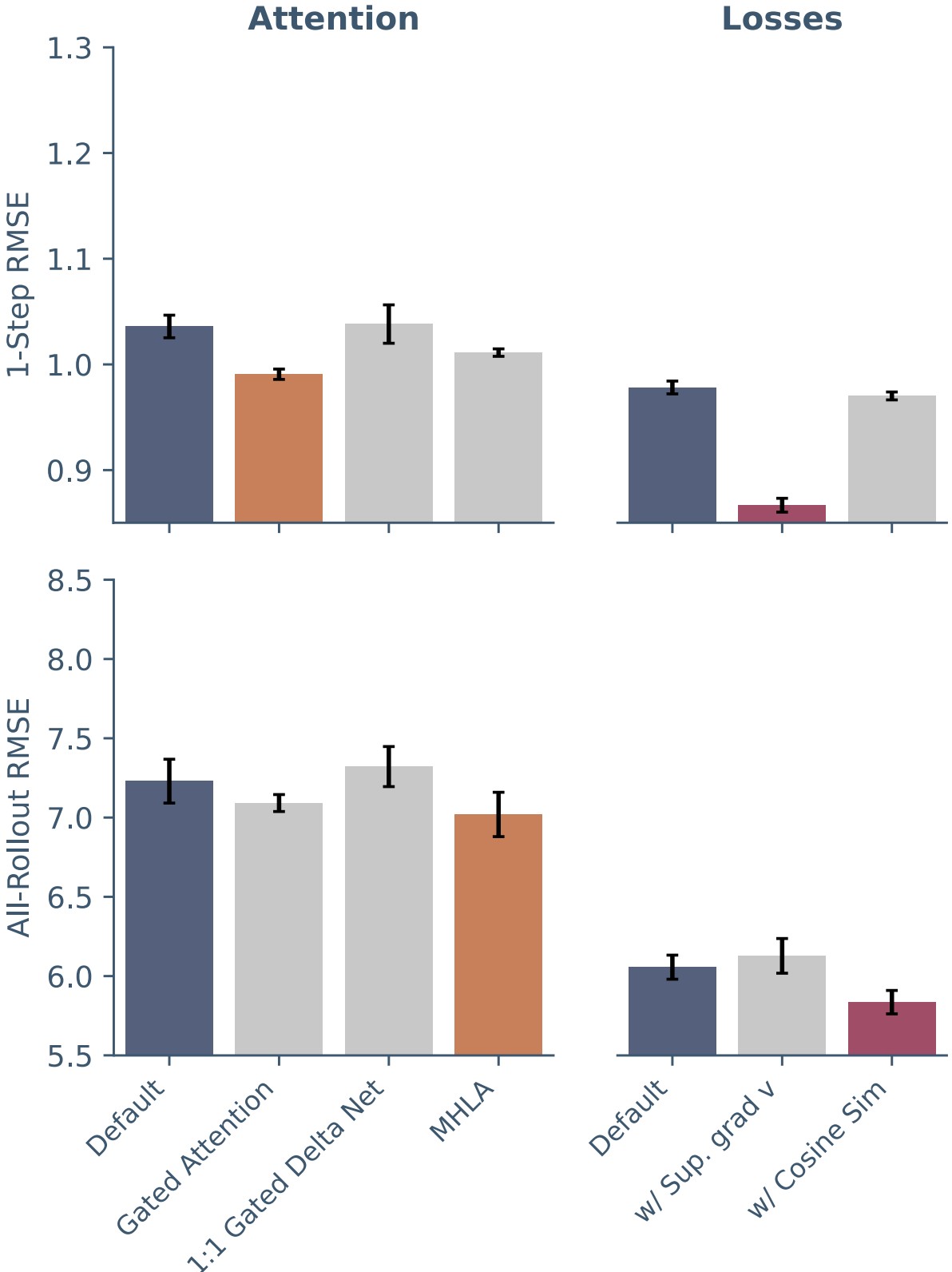

*Figure 10.* **Extra ablation studies.** We present the impact of several variants of attention and of different loss functions.

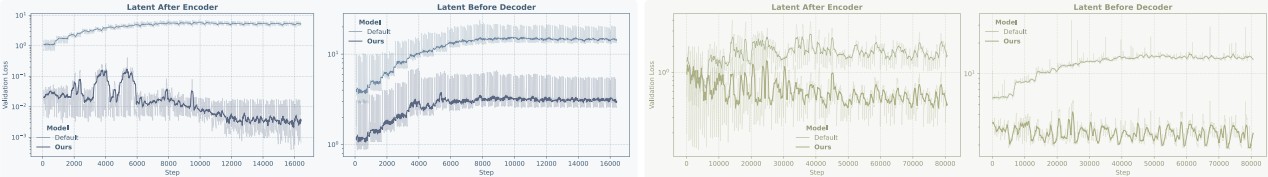

*Figure 11.* **Effect on latent representation.** While training architecture with and without Multi Node Prediction, we encode the next-step target to obtain a latent target, and compute it's difference with the current latent representation at each stage of the architecture. We display the differences after zero spatial processing, and after $L$ spatial processing.

*Figure 12.* The next step velocity (target field) is presented in the top row. The latent representation without MNP is presented in the second row, while the latent representation with MNP is presented in the bottom row.

