# OpenReview forum: "Mesh Based Simulations with Spatial and Temporal awareness"
_ICML.cc/2026/Conference — ICML 2026 regular_

### Official Review · Reviewer_oV8C · 2026-03-07

**Soundness:** 2
**Presentation:** 2
**Significance:** 3
**Originality:** 3
**Overall Recommendation:** 4
**Confidence:** 4

**Summary:**

This paper proposes a unified framework to improve machine-learning-based surrogate models for CFD by revisiting the training paradigm. Specifically, the authors identify limitations of conventional node-wise supervision and explicit time-stepping, and introduce three key components: Multi Node Prediction, Temporal Correction, and Geometric Inductive Biases with 3D RoPE. Experiments on multiple architectures and datasets show consistent gains in accuracy, stability, and long-horizon prediction, with additional evidence of generalization to unseen subtasks.

**Compliance With Llm Reviewing Policy:**

Affirmed.

**Final Justification:**

The authors’ response clarifies the issue regarding dataset usage. The key point is that, for the deforming plate and cylinder datasets, the authors actually used the full set of 1,000 trajectories rather than 100 (which they state was a typo in the paper). They also provided additional information on time step size, included visualization of the results, and discussed the sampling strategy along with supplementary experiments.

These responses address my concerns, and I have therefore increased my score to 4. I hope these revisions and additions will be reflected in the final version of the paper, and that the code will be made publicly available to help advance the field.

**Key Questions For Authors:**

While I have positive attitude towards the method, I have concerns about the experiment details and cannot give a positive score. I would like to see the authors’ response.

- The DeformingPlate and Cylinder datasets in MeshGraphNets each consist of 1,200 trajectories (1,000 for training, 100 for validation, and 100 for testing). In contrast, this paper appears to use only 120 trajectories, and the reason for this choice is not clearly explained. The authors should clarify how these samples were selected and whether the subset was chosen in a way that could favor the reported results. To ensure a fair comparison, the authors are encouraged to provide results under the same dataset settings as MeshGraphNets. As it stands, the large discrepancy in the amount of data used makes the current evaluation less convincing.

- How many frames in each trajectory for these scenarios? This information seems not included in the paper. Also, line 302 says that “the simulation time step ∆*t* are shown in Table 1 and Figure 7”, but I was likely unable to locate this information in either place.

- Although video results may be considered optional, they are in fact highly important for evaluating the dynamic behavior of the simulation, since the task concerns the prediction of temporal dynamics rather than static scenes. In particular, autoregressive errors are often much easier to observe in videos, and any claimed improvements can also be more convincingly demonstrated in this way. However, the paper does not provide any video results, which makes the experimental evaluation less convincing. This is especially important for cases such as the contact regions in the DeformingPlate scenario, where dynamic behavior is critical to assessing performance.

- In the “Multi-Node Prediction” part, what is the sampling strategy? Would the sampled place affect the performance? In another word, would sampling in contact regions and sampling in regions with small deformation yield the same results?

- Typos and writting issues:
  - Figure 1: “Processing Blocs” or “Blocks”?
  - Labels of equations are confused.  For example, labels (2) (3) (4) are for only two equations and labels (11) (12) are for only one equation. This should be fixed for rigor.
  - The paper seems to have formatting issues, with a substantial amount of unnecessary blank space in page 3.

**Limitations:**

While I think this is an interesting approach, the paper would benefit from a more thorough discussion of its potential extension to more realistic physical scenarios like multi-solid systems [1] and etc.

[1] "Unisoma: A Unified Transformer-based Solver for Multi-Solid Systems." ICML 2025.

**Strengths And Weaknesses:**

## Strength
- This paper proposes a unified training framework which is easily implemented and general for most models.
- The authors provide comprehensive experiments to verify the framework’s performance.

## Weakness
See Questions and Limitations.

---

> ### Author Rebuttal · Authors · 2026-03-31
>
> We thank the reviewer for the careful reading and for highlighting both the motivation of the paper and the extensiveness of the experimental study. We appreciate the detailed questions regarding the experimental setup and presentation, and we address them below. Additional figures, experiments and rollout videos are now available in the supplementary webpage: https://sites.google.com/view/mesh-based-simulations-with-sp/home
>
> First and foremost, we apologize for the mistake made in the manuscript regarding the dataset sizes. The Coarse Aneurysm dataset does have 100 training trajectories, **but the Cylinder and DeformingPlate datasets were used entirely, hence having 1000 training trajectories and 100 for testing**. We hope that this can reassure the reviewer regarding its comment on dataset size. We updated l.303-304 in the manuscript regarding this, as well as Table 1 with the summary of the datasets provided below, with the missing number of time steps and size of each time step:
>
> | Dataset           | Solver  | # nodes | Dimension       | # traj | # steps | Δt (s) |
> |------------------|---------|--------:|-----------------|-------:|--------:|-------:|
> | Cylinder         | COMSOL  | 2k      | 2D Fixed Mesh   | 1000   | 600     | 0.01   |
> | Plate            | COMSOL  | 1k      | 3D Fixed Mesh   | 1000   | 400     | -      |
> | Coarse Aneurysm  | Cimlib  | 20k     | 3D Fixed Mesh   | 100    | 80      | 0.01   |
> | Fine Aneurysm    | Cimlib  | 250k    | 3D Fixed Mesh   | 100    | 80      | 0.01   |
>
> We agree with reviewer and **have provided videos** as well as additional figures and tables on top of our rebuttal (see link at the beginning of the rebutal).
>
> We thank the reviewer for its question regarding sampling for the Multi Node strategy, which is a very good one! During our experiments, centers are sampled uniformly inside internal nodes (so excluding all types of boundary nodes). We do not bias sampling toward contact regions or large deformations. We’ve made that clearer in the paper: **l.197 - “Each center is sampled uniformly from internal nodes within the mesh.”**
>
> Indeed, it would have been really interesting to compare the performance of the models depending on where the sampling is biased towards. One could easily imagine better performance in high deformation areas since it would incur a more difficult task. **We followed up on your idea and ran a small experiment on the DeformingPlate dataset, by forcing sampling either close to the contact region, or as far as possible.** Overall, we find no statistically significant differences: both options yield similar improvements as uniform sampling inside the entire mesh. We have updated the manuscript with this ablation: **l.366 - “We’ve also tested different sampling strategies on the DeformingPlate dataset. Centers were sampled either uniformly within the mesh, or biased towards (i) the constraint or (ii) away from it. Overall, the three configurations yield similar results.”**
>
> Finally, we’ve added a small discussion on multi-solid systems and a citation to [1], as well as a correction of the highlighted typos:
>
> - In Figure 1, we replaced blocs by blocks
> - The issues with labels 2,3,4 and 11,12 are both resolved
> - We agree with the unnecessary blank space page 3, which was due to a small typo in a custom latex command. This was resolved as well.
>
> We thank the reviewer again for the constructive comments, which helped improve both the clarity and completeness of the paper. The corrections and additional experiments above directly address the concerns raised.
> If these clarifications resolve the reviewer’s concerns regarding dataset configuration, experimental validation, and methodology details, we would greatly appreciate reconsideration of the score. If any point remains unclear, we would be happy to provide further clarification.

---

> > ### Author Rebuttal · Reviewer_oV8C · 2026-04-03
> >
> > Thanks for the response. I hope all modification,  addition and discussion can be represented in the final version. I will increase my rating. Good Luck!

---

> > > ### Author Response · Authors · 2026-04-07
> > >
> > > We would like to thank the reviewer for the positive feedback on our revisions, and for raising the score.

---

### Official Review · Reviewer_UtpW · 2026-03-07

**Soundness:** 3
**Presentation:** 2
**Significance:** 2
**Originality:** 2
**Overall Recommendation:** 4
**Confidence:** 3

**Summary:**

Overall, the paper proposes three plug-in modifications for mesh-based neural simulators: (i) Multi-Node Prediction (stencil-level supervision instead of node-wise loss), (ii) Temporal Correction (a learned predictor–corrector style update via cross-attention + gating), and (iii) 3D RoPE for attention on meshes. The authors evaluate these components on three backbones (MeshGraphNet, Transformer, Transolver) and report consistent improvements in 1-step and rollout RMSE. The paper’s emphasis on training paradigms (loss and temporal update) rather than only architecture design is valuable.

**Compliance With Llm Reviewing Policy:**

Affirmed.

**Final Justification:**

The authors’ rebuttal adequately addressed my main concerns, so I am comfortable raising my score accordingly.

**Key Questions For Authors:**

1. I would suggest the authors add at least one proper long-range stress-test benchmark (e.g., EAGLE [6] and/or a contact-driven benchmark like HCMT [4]) and include at least one long-range method baseline (e.g., physics-informed rewiring such as PIORF [2]).

**Limitations:**

There were no limitations discussed.

**Strengths And Weaknesses:**

**Strengths**

- Good direction: focusing on supervision and temporal updates is practically relevant and often under-emphasized compared to architecture tweaks.
- The proposed changes are modular and appear easy to adopt.
- The paper includes ablations showing that individual components matter (e.g., cross-attention alone can hurt without gating).

**Weaknesses**
- Related Work is incomplete and causes misleading positioning.
    - The paper discusses higher-order time integration mainly via classical ODE references, but does not cite or discuss modern mesh/graph simulation works that already exploit continuous-time or higher-order integration ideas in learning settings (e.g., FEM-derived continuous-time modeling and solver-based formulations). This weakens the novelty justification and makes some claims read as if the field has not explored these ideas. For example, Finite Element Networks explicitly derive a continuous-time model from FEM and solve an ODE system, which is directly relevant to the “Euler update is the default” narrative [1].
    - Oversquashing is mentioned as a motivation, but introduced without standard citations and without acknowledging recent mesh-specific approaches that address long-range interactions via rewiring or physics-informed graph modifications (e.g., PIORF) [2], or related long-range interaction mechanisms (e.g., Hamiltonian/oscillatory perspectives) [3].
    - Transformer-based mesh dynamics are treated as if “global attention” is the main story, while important transformer lines for mesh dynamics and contact/collision-driven long-range coupling are not discussed (e.g., HCMT) [4]. This is especially problematic because the paper claims improved long-range prediction.
- The paper over-generalizes what “most surrogates” do, and this affects technical credibility.
    - The text states that “most surrogates update the state via an explicit Euler scheme” and later frames “all ML surrogates” as next-step residual predictors. This is not accurate as a blanket statement: there are continuous-time and solver-coupled approaches, and also methods that explicitly decouple a learned spatial operator from a chosen integrator [1].
- Benchmark and baseline selection is not strong enough for the scope of claims.
    - The empirical suite is only three datasets. They are diverse in size (up to ~10k nodes), but they do not convincingly stress the two hardest claims the paper makes: improved long-range dependency handling and stability under challenging regimes. In particular, “long-range stability” is not validated on widely used long-range or turbulence-oriented mesh benchmarks such as EAGLE [6], nor on contact/collision regimes where long-range coupling can be extreme (e.g., HCMT-style settings) [4].

[1] Lienen, M., Günnemann, S. Learning the Dynamics of Physical Systems from Sparse Observations with Finite Element Networks. ICLR 2022.

[2] Yu, Y.-Y., et al. PIORF: Physics-Informed Ollivier-Ricci Flow for Long-Range Interactions in Mesh Graph Neural Networks. ICLR 2025.

[3] Hoang, T., et al. Improving Long-Range Interactions in Graph Neural Simulators via Hamiltonian Dynamics. ICLR 2026.

[4] Yu, Y.-Y., et al. Learning Flexible Body Collision Dynamics with Hierarchical Contact Mesh Transformer (HCMT). ICLR 2024.

[6] Janny, S., et al. EAGLE: Large-scale learning of turbulent fluid dynamics with mesh transformers. ICLR 2023.

---

> ### Author Rebuttal · Authors · 2026-03-31
>
> We thank the reviewer for the careful reading and for highlighting the importance of focusing on training paradigms (supervision and temporal updates) rather than purely architectural changes. We also appreciate the comments regarding related work and benchmark coverage, which helped us improve the paper substantially. Additional figures, experiments and rollout videos are now available in the supplementary webpage: https://sites.google.com/view/mesh-based-simulations-with-sp/home
>
> First, we thank the reviewer for pointing out reference [1]. This is indeed a mistake on our end, and we’ve added it to the camera ready version of the paper as well as tone down some sentences from the paper: **l.62 - “While most approaches use an explicit Euler scheme, some work directly apply Finite Element updates to their surrogate. For example, [1] makes the surrogate estimate how the latent evolves on each mesh’s cells and use those with a fixed Finite Element masses matrix.”**
>
> Second, while we did not mention it in the manuscript, the **Transformer method in our paper does use long-range interaction mechanisms, in the shape of random long distance edges**. However we fully agree that other approaches should have been mentioned. The geometry background of some of the authors helped us shine light on [2], and while we did not have the time to run this as an extra experiment, we have added these precisions to the camera ready version of the paper : **l.163 - \footnote{It’s important to note that the Transformer approach use long range interaction in the form of random jumpers. Other approaches such as PIORF [2] could also be applied here}**
>
> Finally, regarding long range dependency and stability, we’ve run two additional sets of experiments. We ran our improvements on the EAGLE [6] datasets for MeshGraphNet and the Transformer approach. Overall, we find strong improvements for both models.
>
> |Dataset|EAGLE+1|EAGLE+50|EAGLE+250|
> |---|---:|---:|---:|
> |MeshGraphNet|0.0884|0.5615|0.9801|
> |MeshGraphNet + improvements|0.0809|0.5039|0.6913|
> |Transformer|0.0838|0.4122|0.5998|
> |Transformer + improvements|0.0799|0.3528|0.3716|
>
> **We also extended generalization evaluation by testing models with and without our methods on two cases: trained on one cylinder, and then tested on either multiple cylinders, or different shapes.** The rollouts videos are available on the website, and we find our method to improve generalization by a large margin, showing that our upgrades are still relevant when confronted with out-of-distribution field distributions and new flow patterns.
>
> We appreciate that you found the direction valuable and the ablations informative. If the broader literature is acknowledged properly and the claim scope is tightened to match the experiments, we think the main contribution is now better supported thanks to you. If that resolves the novelty and scope concerns, we would be grateful if you would reconsider your grade. If there is still a blocking issue, please point us to it and we will answer it directly.

---

> > ### Author Rebuttal · Reviewer_UtpW · 2026-04-02
> >
> > Thank the authors for the rebuttal and added experiments. My main concerns are fully resolved. I am happy to raise my score accordingly.

---

> > > ### Author Response · Authors · 2026-04-07
> > >
> > > We would like to thank the reviewer for the positive feedback on our revisions, and for raising the score.

---

### Official Review · Reviewer_w1hB · 2026-03-09

**Soundness:** 3
**Presentation:** 3
**Significance:** 2
**Originality:** 2
**Overall Recommendation:** 4
**Confidence:** 4

**Summary:**

This paper proposes a framework that bridges the gap between CFD and geometric deep learning and addresses issues that can be found in traditional methods, such as node-wise predictions and Euler timestepping, which can reduced model's accuracy. To address these issues, they propose Multi-Node Prediction, which changes the training task to predicting the state of a node's entire local topology (target node and its 1-step neighbors), effectively acting as a Sobolev-type regularizer. Additionally, they replace the Euler-like time stepping with a Temporal Correction mechanism, which uses a multi-step Cross-Attention layer that approximates implicit time-stepping. Finally, the framework uses 3D Rotary Positional Embeddings to capture rotational symmetries and anisotropic fluxes in unstructured meshes. The authors evaluated their approach with 3 models: MeshGraphNet, Transolver, and Transformer architectures on 3 datasets. The results show 20% to 30% improvements in rollout accuracy and stability with only a minor increase in computational overhead

**Compliance With Llm Reviewing Policy:**

Affirmed.

**Final Justification:**

This is an interesting paper, and your rebuttal has successfully addressed most of my main concerns regarding scalability, baselines, and physical constraints. I remain positive about this work and have my score of 4 (Weak Accept).

**Key Questions For Authors:**

1. Since the divergence-free loss was omitted due to overfitting, how do you ensure the model doesn't produce unphysical mass or momentum (divergence-free) when predicting on longer rollouts?

2. In many simulations, inaccuracies at the boundary conditions can trigger instabilities or even unrealistic flows. How does the proposed framework handle nodes on the mesh boundary (walls, inlets, outlets) where the local topology is asymmetric? Clarifying if special treatment is required for BCs should be addressed.

3. You validate the framework on only 20 test trajectories per dataset. To what extent can the model generalize to out-of-distribution regimes, such as significantly higher Reynolds numbers or more turbulent flows? In addition, are 20 trajectories sufficient to get the required statistics of the flow?

4. How will the framework scale when applying it to more large-scalle simulations that require millions of cells?

**Limitations:**

A primary concern is that conservation of mass and momentum is not guaranteed. The omission of the divergence-free loss can effectively make the flows unrealistic. Additionally, the paper does not address how the model handles boundary conditions, such as wall functions or inlet/outlet, which are of great importance when dealing with CFD simulations. The framework’s scalability also remains in question when dealing with more realistic CFD simulations, where millions of cells are required. Finally, they only test it on 20 trajectories, which is statistically insufficient to justify the model's reliability, and it remains unclear how error accumulation would behave in more turbulent flows.

**Strengths And Weaknesses:**

# Strengths
### Soundness:
The authors provide detailed proofs and theorems. Specifically proving that the Multi-node Prediction acts as a Sobolev-type regularizer that controls flux errors. As well as, showing that the proposed Temporal Correction mechanism expands the stability region of the surrogate by approximating an implicit theta-method.

### Presentation:
The paper is well-structured. They identify current bottlenecks, such as errors that arise due to node-level predictions and explicit Euler-type stepping of the state-of-the-art algorithms. They also provide both algorithmic pseudocode and figures to explain the framework. Finally, they provide graphs and plots with RMSE for 1-step and All-rollout.

### Significance:
The framework bridges the gap between CFD and Geometric Deep Learning, and couples the ML architectures with the requirements of traditional PDE solvers. That is, the Multi-node predictions and the Temporal Corrections. These implementations improve the rollout accuracy by 20%-30% compared to other state-of-the-art models. In addition, it is interesting to see that MNP helps the model to learn adjacent physics (such as pressure) without being the goal.

### Originality:
Individual components, such as Multi-Token Prediction in NLP or Physics-Informed losses, are existing concepts; the originality is primarily in their integration and the use of Rotary Positional Embeddings.


# Weaknesses
### Soundness:
While the proofs are a highlight, the framework's scalability to larger meshes is not shown. In particular, the largest test case is limited to approximately 10000 nodes. Usually, CFD applications require millions of cells, and the cross-attention mechanism may show a high computational cost at that scale. In addition, the authors state that using a divergence-free loss led to overfitting, which can be seen as a numerical issue, which implies that the approach does not guarantee mass conservation (and/or momentum)

### Presentation:
There is no discussion on how the model handles boundary conditions, such as wall functions or inflow/outflow. In many simulations, inaccuracies at the boundaries and how they are handled can pose global instability. The current paper focuses almost exclusively on internal node predictions.

### Significance:
While the authors claim that the framework is dataset-agnostic, they test it on only 3 specific cases. To make such a justification, it would be interesting to explore also out-of-distribution generalization, where they test flows at higher Re numbers and more turbulent flows. Finally, testing on only 20 trajectories per case can be statistically insufficient for wider use in engineering.

### Originality:
The paper does not sufficiently justify how this specific combination of techniques in the framework outperforms more advanced archtitectures, such as AB-UPT or Transolver++. Without a further comparison to these concurrent works, it is difficult to determine if the proposed framework offers a fundamental advancement or just an incremental improvement of existing "Encode-Process-Decode" approaches

---

> ### Author Rebuttal · Authors · 2026-03-31
>
> We thank the reviewer for the detailed and thoughtful review and appreciate the recognition of both the theoretical contributions and the clarity of the presentation. We address the reviewer’s concerns below and provide additional experiments and rollout videos available at the supplementary webpage: https://sites.google.com/view/mesh-based-simulations-with-sp/home
>
> 1 - We agree that enforcing physical constraints such as mass conservation is an important challenge for learned simulators. However, this limitation is shared by most learned surrogate models, including the baseline architectures evaluated in this work.
>
> Our experiments show that adding a divergence-free residual loss improves the 1-step RMSE but degrades rollout stability due to overfitting, which ultimately increases rollout RMSE (see Appendix Fig. 10). For this reason, we did not retain it in the final framework. We agree this limitation should be stated more clearly and have clarified it in the manuscript **(l.937)  - “On the other hand, this also means that no physical constraints such as mass conservation are ensured to our trajectories.”**
>
> 2 -  We also thank the reviewer for its remark on boundary conditions and agree that this was not stated clearly enough in the manuscript. We used a similar approach as MeshGraphNet for the boundary conditions by assigning a node type to all graph nodes (specified in the appendix l.648), while keeping to a realistic simulation scenario where not all boundary conditions are fully known. Velocity is enforced for all node types except normal and outflow; pressure is only enforced at the inflow nodes. This was precised **l.307 - “Boundary conditions are enforced during the model’s prediction for all node types except Normal and Outflow. For pressure prediction, boundary conditions are only enforced at Inlet nodes.”**
>
> 3 - Regarding the datasets sizes and the 20 evaluation trajectories, we apologize for a mistake made in the manuscript regarding the dataset sizes. The Coarse Aneurysm dataset does have 100 training trajectories, **but the Cylinder and DeformingPlate datasets were used fully (1000 training trajectories and 100 for testing)**. We have updated l.303-304 in the manuscript regarding this issue. We hope that this can reassure the reviewer regarding its comment on framework evaluation.
>
> Furthermore, we agree that it would have been interesting to include experiments regarding our upgrades on unseen cases, **and we extended generalization evaluation by testing models with and without our methods on two cases: trained on one cylinder, and then tested on either multiple cylinders, or different shapes.** We find our method to improve generalization performance by a large margin, rollouts videos for these tests are available on the website. These results demonstrate our improvements when used on unseen geometries and out-of-distribution behaviours such as new vortex and recirculation trajectories. This was added as a last experiment in the paper l.432. While we did not have the time to experiment with larger Reynold’s number datasets, to the best of our knowledge no work has been conducted with similar approaches trying to generalise to a wide range of Reynold’s numbers in a single mixed dataset.
>
> 4 - **We extended our ablations on the fine version of the Aneurysm dataset, where each graph is around 250k nodes on average (and 3 to 4M tetrahedral cells)** to evaluate if our method would scale on larger meshes. We found that (i) our method scales well for larger numbers of nodes, mainly because our MNP strategy focuses on a fixed amount of selected nodes, (ii) inference overhead stays around 10%, and (iii) accuracy improvement remains consistent with our previous experiments. Results are available in the website attached at the beginning of this rebuttal;
>
> To answer the reviewer’s remark regarding the comparison with more advanced architecture, **we applied our methods to a Transolver, Transolver++ and UPT architectures on the Cylinder and DeformingPlate datasets (see figure in the website) as well as the the Transolver and Transolver++ architectures to the Aneurysm dataset.** The results are available below, and were added to the main manuscript by updating the figure 2.
>
> |Architecture|Coarse Aneurysm|Plate|Cylinder|
> |---|---:|---:|---:|
> |MPS|1.42|0.94|3.53|
> |MPS+Ours|1.05|0.82|3.29|
> |Transformer|1.07|1.07|3.3|
> |Transformer+Ours|0.97|0.85|2.90|
> |Transolver|2.2|5.20|3.49|
> |Transolver+Ours|1.6|3.10|3.35|
> |Transolver++|1.59|3.42|3.27|
> |Transolver+++Ours|1.1|2.20|3.00|
> |UPT|—|1.03|3.11|
> |UPT+Ours|—|0.79|2.80|
>
> We thank the reviewer again for the constructive feedback, which helped us improve the paper substantially. The additional experiments above strengthen the claims regarding generalization, scalability, and architectural compatibility.
> If these clarifications address the reviewer’s concerns, we would be grateful if the reviewer would consider revising the score accordingly.

---

> > ### Author Rebuttal · Reviewer_w1hB · 2026-04-03
> >
> > Thanks for the detailed reply from the author. The author's reply addressed most of my concerns. I will keep my score at 4.
> >
> > Some comments:
> > Not having strict conservation laws can pose a fatal flaw, especially for flows such as aneyrisms. A significant amount of research has been conducted in the physics-informed area regarding this exact problem.

---

> > > ### Author Response · Authors · 2026-04-03
> > >
> > > We thank the reviewer for their answer.
> > >
> > > We agree that having no physics-informed losses can lead to flows that, while they appear to have low errors, are inaccurate with respect to physical laws (PDE residuals, conservation laws) or physical quantities (drag, lift, wall shear stress).
> > >
> > > For example, it has now been shown that using a full autoregressive surrogate within an optimization algorithm yields worse results than simpler methods that directly predict the desired quantities, such as drag and lift, for flow control. To that end, we ran two small experiments that we added at the beginning of https://sites.google.com/view/mesh-based-simulations-with-sp/home:
> > >
> > > * Spatial Averaged WSS for all timesteps comparison on 3 test cases from the Aneursym Dataset
> > > * Mass Flow Rate over time at 4 different 2D plans inside the main artery of the aneurysm comparison.
> > >
> > > Overall, our method maintains strong physical accuracy on those two metrics throughout an entire autoregressive rollout, even without being trained with a physics-informed loss.
> > >
> > > We’ve added a small paragraph regarding those experiments at the end of the subsection “5.2. Other Experiments”.
> > >
> > > We also added l939 a small discussion on the use of physics informed loss especially for aneurysm, based on “Physics constrained graph neural network for real time prediction of intracranial aneurysm hemodynamics”, “Fluid–structure interaction analysis of pulsatile flow in arterial aneurysms with physics-informed neural networks and computational fluid dynamics” and “Enhanced vascular flow simulations in aortic aneurysm via physics-informed neural networks and deep operator networks”.
> > >
> > > Finally, as noted earlier, our improvements can be plugged into any architecture, including physics-informed ones. Our goal was to show that our method improved all architectures, regardless of the architecture.
> > >
> > > We hope those final details are sufficient, and thank the reviewer again for their valuable feedback to improve the manuscript!
> > >
> > > If that resolves the novelty and scope concerns, we would be grateful if you would reconsider your grade.

---

### Official Review · Reviewer_YEGH · 2026-03-16

**Soundness:** 2
**Presentation:** 3
**Significance:** 3
**Originality:** 2
**Overall Recommendation:** 4
**Confidence:** 4

**Summary:**

This paper proposes a framework for improving machine-learning surrogates for mesh-based physical simulations by incorporating numerical principles from traditional PDE solvers. The authors argue that current approaches rely on simplistic training paradigms, such as node-wise supervision and explicit Euler-style updates, which ignore spatial consistency and temporal stability in PDE discretizations. Overall, the manuscript's key contribution consists of introducing three modifications to existing mesh-based neural simulators. The authors strive to study an important concept: aligning learning-based simulators with numerical properties of PDE solvers. The proposed method includes (1) Multi-Node Prediction, which supervises predictions for a node and its neighbors to enforce local spatial consistency; (2) Temporal Correction, a predictor–corrector update implemented via cross-attention to improve rollout stability; and (3) 3D rotary positional embeddings to capture orientation-aware spatial relationships on irregular meshes. Experiments on three CFD-style datasets and multiple architectures (MeshGraphNet, Transformer, and Transolver) show improvements in single-step and long-horizon prediction accuracy with modest computational overhead.

**Compliance With Llm Reviewing Policy:**

Affirmed.

**Final Justification:**

My concerns are resolved. I am happy to increase the evaluation accordingly.

**Key Questions For Authors:**

- The experiments primarily evaluate performance on held-out trajectories from the same datasets. How does the proposed framework perform when generalizing to significantly different meshes or geometries?
- The largest meshes used contain roughly 10k nodes. Can the authors comment on how the proposed Multi-Node Prediction and temporal correction modules scale to meshes with hundreds of thousands or millions of nodes?
- How does the proposed framework compare with physics-informed losses (e.g., enforcing PDE residuals or divergence constraints) in terms of stability and generalization?
- Could the authors clarify the exact computational overhead during inference when all components are enabled?

**Limitations:**

yes

**Strengths And Weaknesses:**

### Strengths
- The paper is well written and organized.
- The proposed Multi-Node Prediction (MNP) objective encourages the latent representation of each node to capture local neighborhood structure, which better reflects stencil-based discretizations used in numerical PDE solvers.

### Weakness
- The experiments are conducted on relatively small datasets (≈100 training trajectories) and meshes up to about 10k nodes. It remains unclear whether the proposed approach scales to larger real-world simulations.
- The experiments appear to evaluate models on held-out trajectories from the same datasets. It is unclear how well the approach generalizes to unseen geometries, boundary conditions, or mesh resolutions.
- Several ideas (multi-token prediction, rotary embeddings, predictor–corrector schemes) originate from existing literature. The novelty lies mainly in their combination, but the distinction from closely related approaches could be better articulated.

---

> ### Author Rebuttal · Authors · 2026-03-31
>
> We thank the reviewer for the careful reading and for highlighting the motivation of aligning learning-based simulators with numerical principles. We are pleased that the reviewer appreciated the clarity of the manuscript and the Multi-Node Prediction (MNP) idea. We address the concerns below and provide additional experiments and material (figures, tables, and rollout videos) available at the supplementary webpage: https://sites.google.com/view/mesh-based-simulations-with-sp/home
>
> We would like to clarify an error in the manuscript regarding the dataset description.
> While the Coarse Aneurysm dataset contains 100 training trajectories, **the Cylinder and DeformingPlate datasets were used in their full configuration, namely 1000 training trajectories and 100 test trajectories.**
>
> The text in the manuscript incorrectly stated 100 trajectories in the dataset subsection (l.303). This has now been corrected.
>
> A summary of the datasets is available below:
>
> | Dataset           | Solver  | # nodes | Dimension       | # traj | # steps | Δt (s) |
> |------------------|---------|--------:|-----------------|-------:|--------:|-------:|
> | Cylinder         | COMSOL  | 2k      | 2D Fixed Mesh   | 1000   | 600     | 0.01   |
> | Plate            | COMSOL  | 1k      | 3D Fixed Mesh   | 1000   | 400     | -      |
> | Coarse Aneurysm  | Cimlib  | 20k     | 3D Fixed Mesh   | 100    | 80      | 0.01   |
> | Fine Aneurysm    | Cimlib  | 250k    | 3D Fixed Mesh   | 100    | 80      | 0.01   |
>
> Regarding mesh sizes and scaling, we agree that 10k nodes was quite small compared to realistic problems. **We extended our ablations on the fine version of the Aneurysm dataset, where each graph is around 250k nodes on average (and 3 to 4M tetrahedral elements)**, and obtained the following results:
> - (i) our method scales well even with large numbers of nodes (thanks to the fact that our MNP strategy focuses on a fixed amount of selected nodes),
> - (ii) the inference overhead remains around 10%,
> - and (iii) we find consistent improvement in terms of accuracy. Results are available in the website attached at the beginning of this rebuttal and below:
>
> |Architecture|1-Step Error|All-Rollout Error|
> |---|---:|---:|
> |MeshGraphNet|1.8|14|
> |MeshGraphNet+improvements|0.7|11|
> |Transformer|0.7|9|
> |Transformer+improvements|0.35|5|
>
> These results indicate that the proposed framework remains effective at significantly larger mesh scales.
>
> We agree with the reviewer on the next remark, that the majority of our evaluations are performed on held-out trajectories from the training datasets, as we aligned our approach with evaluation methods in related works.
> We have extended generalization evaluation by testing models with and without our methods on two cases: **trained on one cylinder, and then tested on either multiple cylinders, or different shapes**. The rollouts videos are available on the website, and we find our method to improve generalization by a large margin. These results showcase that our improvements hold when confronted to unseen geometries and out-of-distribution flow patterns such as new vortex and recirculation trajectories. This was added as a last experiment in the paper (l.432).
>
> When it comes to physics-informed loss, we refer the reviewer to figure 10 in the appendix. We experimented with a divergence residual loss (as we work with zero divergence incompressible flows). We found that, while this additional physics-informed loss led to 1-step RMSE improvement, it also induced overfitting and overall increased the rollout RMSE, so we did not keep it for future tests.
>
> Finally, regarding computational overhead, we refer the reviewer to figures 10 in the appendix where we detailed the increase in the number of parameters for our upgrades. Overall, our method creates an increase in computational time of around 10% as mentioned in the Results section (l.327).
>
> We appreciate the reviewer’s constructive feedback regarding scaling and generalization. The additional experiments above strengthen the paper and confirm that the proposed framework remains effective on larger meshes and under geometry shifts.
> If these clarifications address the reviewer’s concerns, we would be grateful if the reviewer would consider revising the score accordingly.

---

> > ### Author Rebuttal · Reviewer_YEGH · 2026-04-07
> >
> > Thanks for the reply. My concerns are resolved. I am happy to increase the evaluation accordingly.

---

> > > ### Author Response · Authors · 2026-04-07
> > >
> > > We would like to thank the reviewer for the positive feedback on our revisions, and for raising the score.

---

### Decision · Program_Chairs · 2026-04-30

**Decision:**

Accept (regular)

**Comment:**

This paper initially received mixed recommendations from the reviewers. Reviewers raised questions about the paper's experiments with relatively small data sizes, the novelty of the idea relative to prior art, the method's technical soundness, and its benchmark and baseline choices. The rebuttal and the reviewer-author discussion were constructive, resolving most of the concerns raised in the initial review. After the post-rebuttal discussion, all reviewers have converged on weak acceptance. Therefore, I am recommending that ICML accept this work.